# Assessment of seasonal soil moisture forecasts over Central Mediterranean

Lorenzo Silvestri[1], Miriam Saraceni[2], Bruno Brunone[3], Silvia Meniconi[3], Giulia Passadore[4], and Paolina Bongioannini Cerlini[5]

[1]Department of Engineering Enzo Ferrari, DIEF, University of Modena and Reggio Emilia, Modena, Italy
[2]Interuniversity Research Center, CIRIAF, University of Perugia, Perugia, Italy
[3]Department of Civil and Environmental Engineering, DICA, University of Perugia, Italy
[4]Department of Civil, Environmental and Architectural Engineering, ICEA, University of Padova, Italy
[5]Department of Physics and Geology, FIS-GEO, University of Perugia, Italy

**Correspondence:** Lorenzo Silvestri (lorenzo.silvestri@unimore.it)

**Abstract.** It is highly likely that in the next future the Mediterranean region will experience increased aridity and hydrological droughts. Therefore, seasonal forecasts of soil moisture can be a valuable resource for agriculture and for evaluating the flux in the vadose zone towards shallow unconfined aquifers. However, their accuracy in this region has not been evaluated against observations. This study presents an evaluation of soil moisture in the Central Mediterranean region during the period 2001-2021 using the seasonal forecast system (SEAS5) of the European Center for Medium-range Weather Forecast (ECMWF). In this perspective, standardized anomalies of soil moisture are compared with observed values in ERA5-LAND reanalysis of ECMWF. In terms of the average magnitude of the forecast error and the anomaly correlation coefficient, the forecasts demonstrate good performance only in certain regions of the domain for the deepest soil layer: Hungary, Peninsular Italy, Internal areas of Balkan Peninsula, Provence, Sardinia, and Sicily. These regions correspond to those with the largest memory time scale of soil moisture, which do not exhibit a complex orography. The obtained results show that seasonal forecasts are useful to detect wet and dry events for the deepest soil layer in the mentioned regions, with lead-times of up to six months. In these regions, the area under the Relative Operating Characteristic (ROC) curve can reach values larger than 0.8. For all soil layers, dry events are generally better captured than wet events; the best forecast skill, on average, is obtained for the events where antecedent condition is correspondent to the present condition (dry after dry, wet after wet). To illustrate these features, the case study of the 2012 drought period demonstrates the capacity of the SEAS5 model to forecast such an event for Central and Northern Italy with a six-month lead time. Furthermore, the close correlation between soil moisture and the observed water table in shallow unconfined aquifers in Italy underscores the significant potential of seasonal soil moisture forecasts for underground water management applications.

## 1 Introduction

Soil moisture, starting from the terrestrial surface to the deepest soil layers, represents an invaluable parameter which has a fundamental role in the dynamics of the earth system (McColl et al., 2017). Its variability results from the complex interaction between the atmosphere, vegetation and soil processes.

On the terrestrial surface, soil moisture is an essential component of the Earth surface energy budget, influencing the surface heat fluxes and evapotranspiration from land to atmosphere (Seneviratne et al., 2010). From the climate point of view, Mueller and Seneviratne (2012) showed that the number of hot days is largely determined by a precipitation deficit and, as a consequence, by small values of soil moisture. This coupling between atmospheric temperature and soil moisture is usually defined as soil moisture-temperature feedback, where drier soils determine warmer atmosphere (Seneviratne et al., 2010). Such feedback has the potential to exacerbate global warming by altering the surface heat balance (Qiao et al., 2023). Other studies (Hohenegger et al., 2009; Hohenegger and Stevens, 2018; Taylor, 2008; Taylor et al., 2010) concentrated on the reciprocal influence between soil moisture and precipitation, which is referred to as the soil moisture-precipitation feedback. A number of processes may contribute to this feedback, acting both on a synoptic scale (by modifying synoptic settings and enhancing the large-scale transport of water vapor) and locally (by modifying boundary layer characteristics and influencing the organization of convection). Nevertheless, it remains challenging to ascertain an overall sign (positive or negative) for this feedback.

The soil moisture available in the root zone is essential for vegetation and agriculture. Its values can be used as indexes for detecting hydrological drought (Spennemann and Saulo, 2015). Through its impact on photosynthesis processes, Humphrey et al. (2021) found that the variability of soil moisture in climate model simulations drive the 90% of the inter-annual variability of the global land carbon uptake.

The deep soil moisture is a fundamental feature with respect to the flux in the vadose zone towards shallow unconfined aquifers. For example, Rodell et al. (2007) used the satellite observed terrestrial water storage from the Gravity Recovery and Climate Experiment (GRACE) to determine the groundwater storage. Later, Getirana et al. (2020) demonstrated that the initialization of seasonal forecast with such data improves groundwater forecasts in the USA. In addition, Li et al. (2021) evaluated groundwater recharge from different land surface models and found that the seasonal cycle of simulated groundwater storage compared well with in situ groundwater observations.

Despite its fundamental role, in situ observations of soil moisture are scarce. Satellite and reanalysis products can provide a useful alternative to fill this gap. However, direct satellite observations are possible only for the first few centimeters below the surface (Dorigo et al., 2021). These surface observations can be propagated through the root zone by filtering operations, empirical models or land surface models. Reanalyses offer a great alternative for studying soil moisture and they are characterized by significant correlations with in situ observations. Li et al. (2020) compared different reanalysis and found ERA5, the fifth generation reanalysis of the European Center for Medium range Weather Foreacasts (ECMWF), to show the highest skill. Also Bongioannini Cerlini et al. (2017, 2021) showed the strong correlation between ERA5 flux and aquifer water table observations. The same was found by Spennemann and Saulo (2015) between the Global Land Data Assimilation System (GLDAS)

and multi-satellite soil moisture anomalies. The utility of soil moisture data from land surface models employed within atmospheric general circulation models hinges not on the soil moisture value itself, but on its temporal variations, which are particularly well represented when compared to observations (Koster et al., 2009). By analyzing different reanalysis and land surface models with respect to observational data in Central Italy, Bongioannini Cerlini et al. (2023) found, on average, the best performances of the ERA5 reanalysis with respect to other well-established reanalysis. As a further feature suggesting the use of ERA5, its good performance in terms of water budget evaluation in closed lakes must be mentioned (Bongioannini Cerlini et al., 2022; Saraceni et al., 2024). For these reasons, in this paper ERA5 reanalysis, and its land component ERA5-LAND (Muñoz-Sabater et al., 2021), will be used as a reference soil moisture condition.

There is high confidence that the Mediterranean region will suffer from a larger aridity and an increase in hydrological droughts (Ranasinghe et al., 2021). Moreover, aridity can heavily impact the snowmelt recharge of the aquifers in the mountain ranges of the Mediterranean area, further affecting hydrological droughts (Lorenzi et al., 2024; Doummar et al., 2018) as well as vegetation phenology (Cerlini et al., 2022). In this context of climate change, sub-seasonal to seasonal (S2S) forecasts are a fundamental tool for adaptation strategies, especially regarding water resources management. The accuracy of S2S forecast system relies on the simulation of the response of the atmosphere to the slowly varying states of the ocean and land surface (Koster et al., 2004). Johnson et al. (2019) demonstrates how SEAS5, the seasonal forecasting system of ECMWF, has a particular strength in the prediction of El Niño Southern Oscillation (ENSO). de Boisséson and Balmaseda (2024) found globally useful forecast skill when predicting the occurrence of marine heatwaves (prolonged period of extremely warm sea surface temperature) for the two seasons after the forecast initialization date. Crespi et al. (2021) analyze the forecast skill of SEAS5 for three key climate variables (temperature, precipitation and wind speed) over Europe and found such forecasts useful for climate services after a proper bias-adjustment method was applied. Prodhomme et al. (2021) found that seasonal forecasts from the SEAS5 system starting from the early May can provide useful information about the probability of occurrence of European summer heatwaves. A recent study over the Mediterranean region by Calì Quaglia et al. (2022) found that individual seasonal forecasting systems outperform elementary forecasts of precipitation anomalies based on persistence or climatology. However, the added value is not uniform over the Mediterranean area. The same dis-homogeneity and potential usefulness of seasonal forecast in the Mediterranean area was found also by Costa-Saura et al. (2022) for agriculture and forestry. However, the same analysis could bring different results in regions with marked orographic impact and land-sea contrast such as the Mediterranean region. Ceglar and Toreti (2021) show that seasonal climate forecast by SEAS5 provides useful information for decision-making processes in the European winter wheat-producing sector, by analyzing minimum and maximum daily temperature and daily total precipitation. In particular, drought events were better predicted than excessive wetness periods.

On the scale of S2S forecasts, soil moisture is one of the most impactful land parameter and is crucial for the forecast skill (Koster et al., 2004, 2016; Merryfield et al., 2020; Dirmeyer et al., 2018). Esit et al. (2021) found that land initialization contributes to approximately a third of the total soil moisture predictability, while the remaining part is attributable to ocean conditions. Moreover, they found that the same initialization can provide limited skill in the precipitation forecast but enough skill in the soil moisture forecast. This result suggests that skillful seasonal prediction can be made on drought occurrence

focusing on the soil state. This can be attributed to reduced variability of soil moisture which is an order of magnitude smaller than that of rainfall. The study by Kumar et al. (2019) in North America suggested that this source of predictability is connected to the soil moisture reemergence process, in which moisture anomalies stored in the deep soil layer can "reemerge" to the surface, restoring the earlier root zone anomaly and providing a year-to-year soil moisture memory. Spennemann et al. (2017) found that seasonal forecast of Standardized Soil Moisture Anomalies (SSMA) perform better than forecast of precipitation by using the CFSv2 (Climate Forecast System) in South America. Moreover, the performance were found to be higher for austral winter than summer, and for dry events rather than wet episodes. This result shows the value of seasonal forecast of SSMA for their use for agricultural drought monitoring. A recent study by Boas et al. (2023) found that the Community Land Model (CLM5), forced by SEAS5 seasonal forecasts, satisfactorily reproduces the inter-annual variation of crop yield and also the high- and low-yield seasons in Germany and Australia. However, a systematic bias of soil moisture was found when comparing with satellite observations.

Most of the above results apply to large continental regions in North and South America, while in Europe seasonal forecast performances are mostly evaluated for surface atmospheric variables. Accordingly, to fill this gap, this paper focuses on evaluating seasonal forecasts of soil moisture for water resources management, with particular attention to wet and dry events. The key questions addressed in this study are: i) can the seasonal forecast over the central Mediterranean be used to predict the soil moisture behavior? ii) does performance vary depending on whether a forecast period is dry or wet? iii) can we use such information to develop a real-time applications for water resource management?

The paper is structured as it follows. Section 2 describes the study area, the seasonal forecast system and the reanalysis data used to validate the forecast. Section 3 provides a description of methods for evaluating the forecast performance. Results are reported in Section 4, while Section 5 examines some case studies of extreme dry and wet periods. Finally, section 6 summarizes and discuss the main findings of this study.

## 2 Study area and data

### 2.1 Study area

This study focuses on the Central part of the Mediterranean region ($5^oE$-$25^oE$, $35^oN$-$50^oN$), as shown in Figure 1. Such an area represents a challenge for seasonal forecasts (Doblas-Reyes et al., 2013) for different reasons. First it is greatly influenced by climate change, sometimes recognized as a hot spot. As stated by the sixth IPCC report (Ranasinghe et al., 2021), in the Mediterranean region there is a strong agreement between regional climate models that precipitation will decrease and temperature will increase by mid- and end-century for the Representative Concentration Pathway (RCP-8.5) and the Shared Socioeconomic Pathways (SSP5-8.5) scenarios. Therefore, with high confidence, this area will suffer from a larger aridity and an increase in hydrological droughts. Second, the complex orography of this region (the Alps, the Apennines, the Dinaric Alps, and part of the Atlas mountains) complicates the precipitation forecasts (Silvestri et al., 2022). Finally, additional sources of uncertainties comes from land-sea contrast, atmosphere-sea interactions, and the complex dynamics of extra-tropical atmo-

spheric circulation.

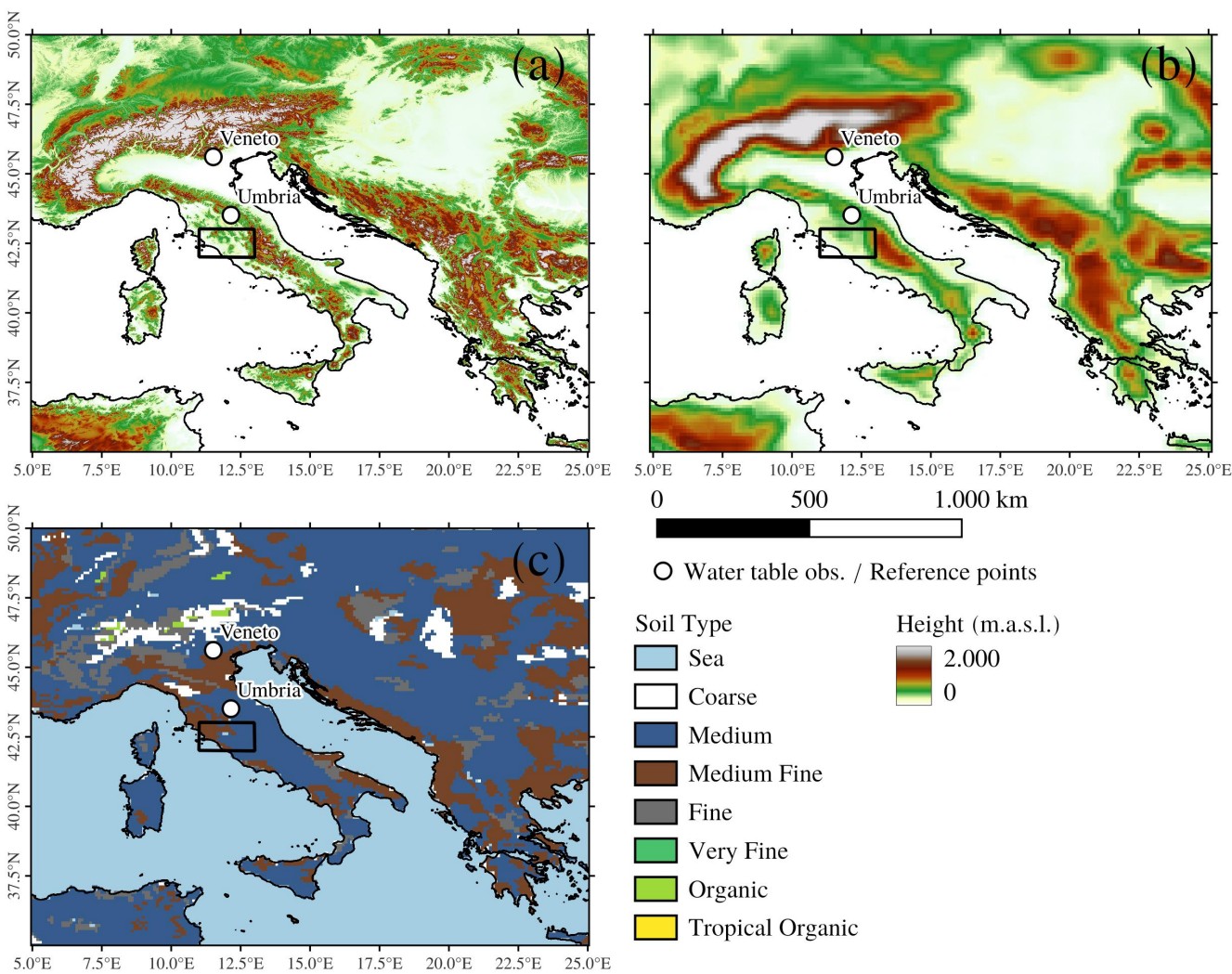

**Figure 1.** The study area and its orography as represented by: (a) Digital Elevation Model with 1 km resolution (Danielson et al., 2011, GMTED) ; (b) ERA5 reanalysis with horizontal resolution of about 31 km (which can be taken as a reference also for SEAS5 system which has a resolution of about 36 km); (c) is the soil type categories as represented in ERA5-LAND. White dots represent water table observations in the Veneto and Umbria regions analyzed in this paper as case studies. The black rectangular area is used as a reference area for averaging anomaly correlation coefficients in Central Italy.

## 2.2 Soil moisture reanalysis

ERA5-LAND (Muñoz-Sabater et al., 2021) and ERA5 reanalysis itself (Hersbach et al., 2020) are used here as a reference dataset for soil moisture since it has been shown to have good performance in representing the observed soil moisture (Muñoz-Sabater et al., 2021; Li et al., 2020), especially regarding its seasonal cycle.

ERA5 is produced by using the Integrated Forecasting System (IFS) model version CY42R1. The land surface model is HT-ESSEL (Balsamo et al., 2009) which interacts directly with the atmosphere. Soil moisture is a prognostic variable and, for this reason, its initial value is needed to run the model. Precisely, observations in ERA5 are assimilated each 12 hours through a 4d-variational (4d-Var) approach. A simplified Extended Kalman Filter (De Rosnay et al., 2013) is implemented in IFS to produce the initial condition for the soil moisture analysis. It is based on two different sources of observations (Albergel et al., 2012): the surface observations of temperature and relative humidity from synoptic stations (SYNOP) measured at 2 m above the ground level (the so-called screen level), and MetOp-A, MetOp-B Advanced Scatterometer (ASCAT) soil moisture data from satellites. Screen-level parameters are indirectly related to soil moisture, while satellites provide a more direct measurement of the surface soil moisture. Since the latter source is capable of describing only the top few centimeters of the soil (Albergel et al., 2012), the root-zone soil moisture is estimated by propagating downwards this information by means of the H-TESSEL hydrological model.

The high horizontal resolution of ERA5 ($0.28^o \approx 31$ km), together with an improved physics and data assimilation methods, make this reanalysis one of the most reliable and physically consistent dataset of global soil moisture. Seasonal forecasts products from SEAS5 come from a different model version, with different initial conditions, different data assimilation methods, and different horizontal resolution (see Johnson et al., 2019, for more details).

ERA5-LAND, the land component of ERA5, is produced by running the H-TESSEL hydrological model at a higher horizontal resolution of 9 km. The static and climatological fields, like soil type, land-sea mask and orography, are the same as ERA5 but interpolated to a higher resolution grid. Soil type, which is a relevant parameter for calculating soil moisture, is shown in Figure 1c. When moving across different grids, the dominant soil type is selected in order to preserve hydraulic properties (Balsamo et al., 2009). This is true also for the seasonal forecast system SEAS5 (see below). The other difference between ERA5 and ERA5-LAND is the thermodynamic input. In particular, in ERA5-LAND the surface pressure and the temperature are adjusted for the altitude through a daily environmental lapse rate obtained by ERA5 data. As discussed in Muñoz-Sabater et al. (2021), such a dynamical downscaling of ERA5 implies consistent improvements for soil moisture especially in the root zone, when compared to soil moisture observations. Instead, for the top layer, ERA5-LAND slightly improves the ERA5 estimates. The main reasons behind such improvements are due to a better representation of the soil type which changes the saturation level of soil moisture, thus affecting evapotranspiration.

## 2.3 The seasonal forecasting system (SEAS5)

Seasonal forecasts of monthly mean soil moisture were taken from the fifth generation seasonal forecasting system (SEAS5) of ECMWF (Johnson et al., 2019). In the following, we briefly provide a few details on SEAS5, but the reader is referred to Johnson et al. (2019) for further information.

SEAS5 is based on cycle 43r1 of the Integrated Forecast System (IFS) and consists of a coupled system of atmospheric, land surface, oceanic, and sea-ice components. The horizontal resolution of the atmospheric model physics is about 36 km (O320 grid) with 91 levels in the vertical. The ocean model is ORCA ($0.25^o$) with 75 levels in the vertical. Land surface is represented through the H-TESSEL model (Balsamo et al., 2009), while sea-ice is treated with the LIM2 model (Fichefet and Maqueda, 1997). The atmosphere and land surface are initialized using ECMWF operational analyses, while the ocean and sea-ice are initialized using OCEAN5 (Zuo et al., 2019), which combines the ORAS5 historical ocean reanalysis with the OCEAN5-RT daily ocean analysis.

In this paper, SEAS5 hindcasts (or reforecasts, that is forecasts produced for the past period between 2001-2016) and forecasts between 2016-2021, for a total period of twenty years (2001-2021), are used. There is no substantial difference between the system set up for hindcasts (reforecasts) and forecasts. Such a distinction is done since the SEAS5 system become operational in 2017 and the actual forecasts were started from that period. Hindcasts are performed in order to extend the available time period of seasonal forecasts and allow a better calibration. Moreover, the period until 2016 is used as a reference period for calculating anomalies and the bias adjustment of forecasts with respect to observations. Each forecast consists of different members and lead-time months. The SEAS5 reforecasts have 25 members, while the forecasts have 51 members. To have a homogeneous number of members throughout all the analyzed period, only the first 25 forecast members are considered. Regarding the lead-times, each forecast consists of 7-month time steps and it is initialized at the beginning of each month. In our analysis, all lead-times spanning from 1 to 6 months are considered.

## 2.4 Water table observations

In this study, we use surface observations of water table as a direct proxy for dry and wet case study events. We select 2 piezometers in two different Italian regions, Umbria and Veneto, respectively located in the Central and Northern part of Italy (white dots in Figure 1). The piezometers monitor two different shallow alluvial and unconfined aquifers with a mean depth of water table below 10 m, whose evolution has been found to be representative of a large area surrounding the point observation (Bongioannini Cerlini et al., 2021). In such unconfined aquifers, the flux in the vadose zone is the result of the direct interaction between land and atmosphere.

The measurements of the water table elevation are provided by the regional piezometric network of the Umbria region, managed by the Regional Environmental Protection Agency [Agenzia Regionale per la Protezione Ambientale (ARPA)] and by local water management services in Veneto. Daily water table data have been collected for at last 10 years and subjected to preliminary quality control procedures (see Bongioannini Cerlini et al., 2021, for a detailed description of the quality controls),

before calculating their monthly mean and the corresponding standardized anomalies.

## 3  Methods

Monthly mean values of the soil moisture, $\theta$, from seasonal forecasts are validated against monthly mean values of soil moisture
from both ERA5 and ERA5-LAND reanalysis. Both datasets are interpolated over a regular grid of $0.125^o$ of horizontal
resolution. The number and the depth of soil layers in each column is the same in both SEAS5, ERA5 and ERA5-LAND: four
soil layers at a depth of 7 cm (soil layer 1), 28 cm (soil layer 2), 100 cm (soil layer 3), and 289 cm (soil layer 4), respectively. The
soil type, when passing across different grids, is taken as the prevailing soil type in order to preserve soil hydraulic properties
(Balsamo et al., 2009).

The evaluation of seasonal forecasts and also the discrimination of dry and wet periods is performed over the Standardized Soil
Moisture Anomaly (SSMA). Following the approach by Spennemann et al. (2017), SSMA is calculated at each grid point (i,j),
month (m, from January to December), year (y) and soil layer (k) as:

$$SSMA_k(i,j,m,y) = \left(\theta_k(i,j,m,y) - \overline{\theta_k}(i,j,m)\right)/\sigma_{\theta_k}(i,j,m) \tag{1}$$

where $\bar{} $ = time average operator over all the reference year, and $\sigma$ = standard deviation operator. The time period considered
for the forecast validation spans 20 years from 2001 to 2021, while the reference time period considered for evaluating the
monthly climatology and standard deviation ranges from 2001 to 2016.

The same reference period is also considered for the bias adjustment of seasonal forecast. The method used in this work is the
simple Mean and Variance Adjustment method as described by Manzanas et al. (2019). Each member mean and variance over
each grid point is bias-adjusted with respect to the ERA5 observation mean and variance over the period 2001-2016, in the
following form:

$$\theta_k'(l,m,y,n) = \left(\theta_k(l,m,y,n,) - \widehat{\theta_k}(l,m)\right)\frac{\sigma_{obs}(m)}{\sigma_f(l,m)} + \overline{\theta_k^{obs}}(m) \tag{2}$$

where $l$ is the forecast lead time, $m$ is the forecast month, $n$ is the index representing each ensemble member, $\hat{\theta}_k(l,m)$ is
the ensemble and time average of forecasts for each lead time and month over the reference period, $\sigma_f(l,m)$ is the standard
deviation of the complete ensemble for each lead time and month over the reference period, $\overline{\theta_k^{obs}}$ is the time average of all
observation for the considered month over the reference period, and $\sigma_{obs}(m)$ is the standard deviation of all observations for
the considered month over the reference period. The bias adjustment is computed for each forecast lead-time ($l$, from 1 to 6
months). In this way, the bias and variance adjustment take into account both the forecast month and the forecast lead time,
which has been found to be beneficial in previous work by Kumar et al. (2014). Although the simplest among different meth-
ods, Manzanas et al. (2019) demonstrated that MVA methods represent a good compromise between computational cost and
performance. This is particularly relevant, since the final aim of this study is to develop real-time applications for climate ser-
vices. The final effect of the bias adjustment on the forecast ensemble mean is shown in Figure 2, where the Umbria reference

grid point (see Figure 1) is shown as an example.

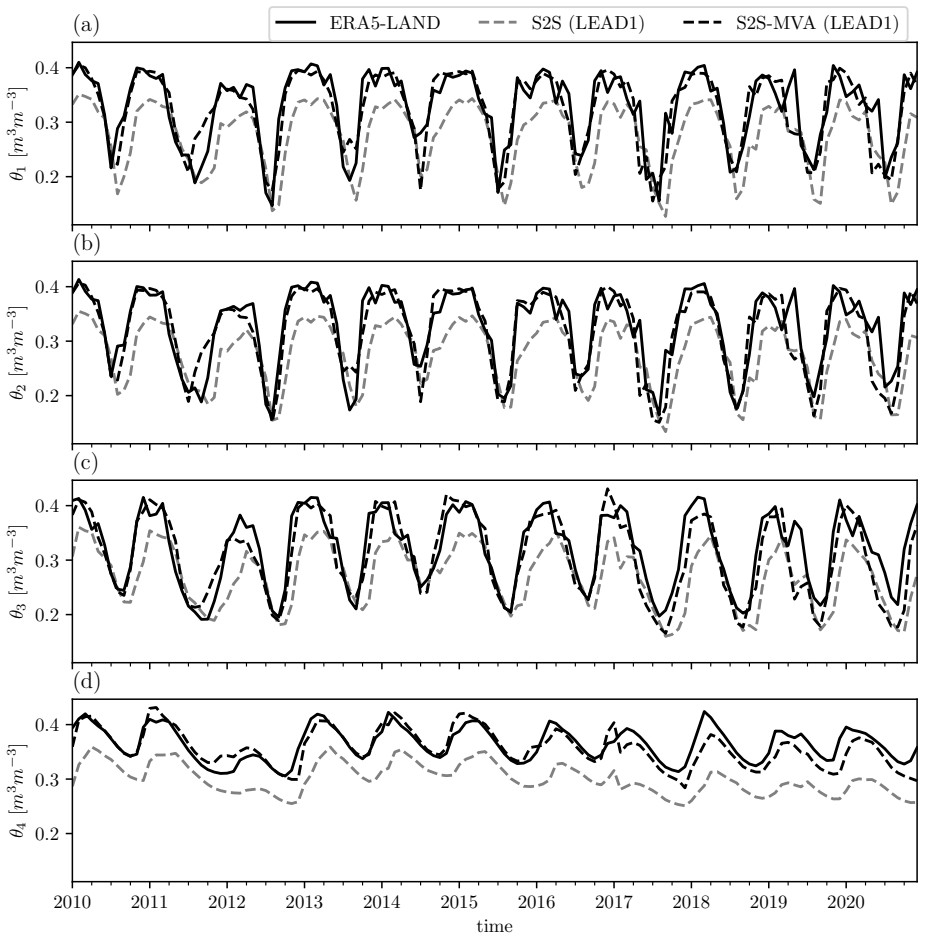

**Figure 2.** Soil moisture time series over Umbria for (a) soil layer 1, (b) soil layer 2, (c) soil layer 3, (d) soil layer 4. Different lines represent ERA5-LAND reanalysis (black solid line), SEAS5 seasonal forecast without bias adjustment at lead time 1 month (gray dashed line), SEAS5 seasonal forecast with mean and variance bias adjustment at lead time 1 month (SEAS5-MVA, black dashed line).

In order to analyze the variability of soil moisture and to compare it across different soil layers, we compute the memory
time-scale of each layer as the e-folding time of the temporal autocorrelation function. The autocorrelation is evaluated by calculating the Spearman correlation coefficient, shifting the time series by a temporal lag which is between 0 and 365 days. The corresponding time when the correlation coefficient becomes lower than $e^{-1}$ is taken as the memory time scale of that grid point and soil layer. This time scale is evaluated by considering ERA-LAND daily mean soil moisture data over all the domain. An example of this procedure for the Umbria reference point is reported in Figure 3. As expected, the deeper the soil
layer the longer the memory time scale. This behavior can be observed for all grid points of the study domain, as it will be

shown later in subsection 4.1. A further interesting feature is pointed out by the autocorrelation structure. Precisely, after an initial decay (as expected), the autocorrelation shows a rebound with a secondary statistically significant maximum at a lag of approximately 300-350 days. Such rebound could be indicative either of the seasonal cycle or of the reemergence of soil moisture anomalies as hypothesized by (Kumar et al., 2019). However this behavior is not representative of all regions and then merits further explorations in future research.

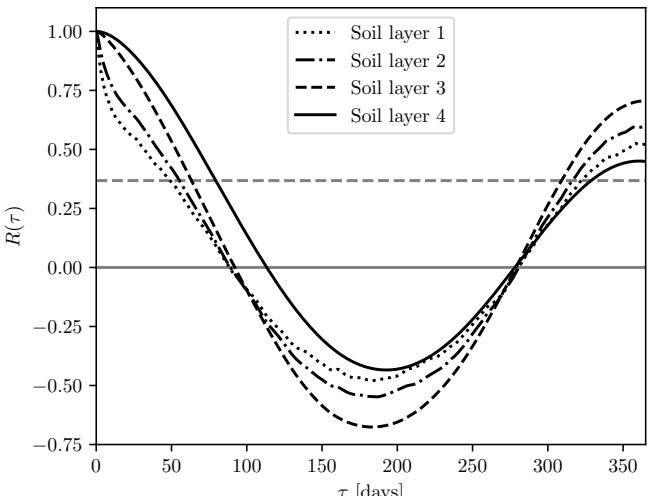

**Figure 3.** Memory time scale over Umbria region for the different soil layers: soil layer 1 (dotted line), soil layer 2 (dashed-dotted line), soil layer 3 (dashed line), soil layer 4 (solid line). The temporal correlation refers to the time series shown in Figure 2. The dashed gray line represents the threshold $e^{-1}$ and the corresponding e-folding time (the time when correlation is lower than this threshold) represents the memory time scale of each soil layer.

The performance of SSMA forecasts is evaluated through three different metrics, two deterministic and one probabilistic. First, the average magnitude error of SSMA ensemble mean is evaluated trough the Root-Mean-Squared Error (RMSE). This metric, by definition, puts greater influence on large errors than smaller errors. RMSE is commonly used both in weather forecast performance assessment (Robertson et al., 2015; Johnson et al., 2019) and seasonal streamflow forecasting (Mendoza et al., 2017; Yuan, 2016). Successively, the Anomaly Correlation Coefficient (ACC) is used to measure the correspondence between forecasted and observed ensemble mean SSMA. ACC is the most widely used skill metric for evaluating the skill of deterministic forecast (Doblas-Reyes et al., 2013; Mishra et al., 2019; Johnson et al., 2019; Costa-Saura et al., 2022) and it is not sensitive to forecast bias. Then the ability of SEAS5 ensemble system to discriminate between different event type is measured by the area under the Relative Operating Characteristic (ROC) curve (Wilks, 2011; Madrigal et al., 2018; Carrao et al., 2018). An example of the procedure for the evaluation of the ROC curve is reported in Figure 4 for the Umbria reference point. In particular, dry and wet events have been defined as those with the SSMA being smaller or larger than 1, respectively (see Figure 4a). An ensemble (probabilistic) forecast will have a certain probability of detecting that event. Using a set of

increasing probability thresholds, we build a contingency table (true and false positive, true and false negative). Then we calculate the true positive rate (or probability of detection) and the false positive rate (or false alarm rate) for each probability bin. The ROC curve is obtained by plotting the true positive rate against the false positive rate as shown in Figure 4b for the different probability bin. For each probability bin, the true positive rate should be larger than the false alarm rate, otherwise the forecast is not useful. Therefore, the area under the ROC curve can be used as a score to evaluate the usefulness of a forecast. The diagonal line on Figure 4b, indicates no skill (ROC area close to 0.5), while the perfect forecast would have a ROC area equal to 1. Each metric has been evaluated for the different forecast lead-times. The first two metrics (RMSE and ACC) are evaluated by considering the ensemble mean SSMA values, while the latter (ROC) is evaluated by considering all the ensemble members. All metrics calculations rely on the xskillscore Python Package (https://github.com/xarray-contrib/xskillscore).

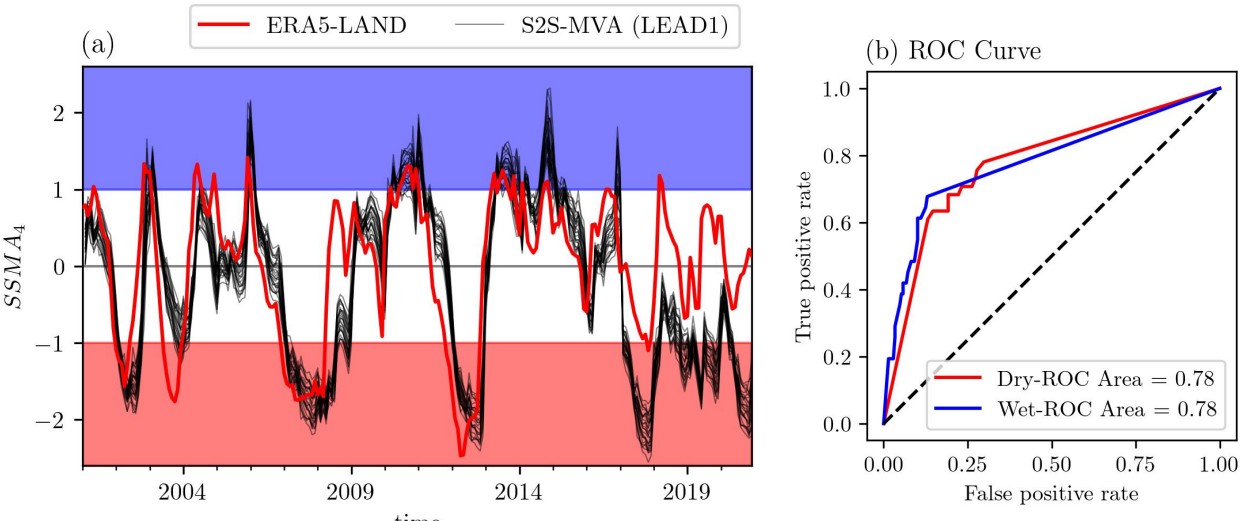

**Figure 4.** Example of estimation of ROC curve over Umbria (white dot in Figure 1): (a) the time series of SSMA4 over soil layer 4 as observed by ERA5-LAND (red line) and forecasted by the 25 members of S2S bias-corrected ensemble at lead-time 1 months(S2S-MVA (LEAD1), black lines). Dry (SSMA4 < 1) and wet (SSMA4 > 1) events are highlighted by red and blue shading, respectively; and (b) ROC curve for the time series shown in (a) for dry events (red line) and wet events (blue line). The value of the area under the ROC curve is reported in the legend.

## 4 Results

In this section, the obtained results are analyzed in terms of soil moisture variability and forecast performance metrics.

## 4.1 Soil moisture variability

The monthly mean soil moisture variability for each soil layer is shown in Figure 5. Both ERA5 and ERA5-LAND datasets are reported in order to highlight the differences between these datasets. Along with to the soil moisture reanalysis products, also the unbiased version of SEAS5 is reported at the following lead times: 1 month (L1), 3 months (L3) and six months (L6). The boxplot represents the spread of soil moisture both in time (i.e., soil moisture variations across different years for the same month) and space (i.e., soil moisture variations across all the domain grid points).

Figure 5 shows that there is a strong seasonal cycle in the upper layers (Figures 5a-c) and a weak seasonal cycle in the deepest soil layer (Figure 5d), where the median values of soil moisture exhibit very small variations across the year. Regarding the median values, ERA5 differs from ERA5-LAND. In particular, ERA5 has smaller values of soil moisture with respect to ERA5-LAND. This may be related to the different thermodynamic input and soil properties, which modify the evapotranspiration contribution, as discussed in Muñoz-Sabater et al. (2021). The need for a bias-adjustment of the seasonal forecast is evident in Figure 5: the median values are not always aligned with the soil moisture reanalysis, especially during the autumn season (September, October, November) for the three uppermost soil layers. In the deepest soil layer the bias is smaller and homogeneous throughout the year, but still present.

The spread of soil moisture across all domain points and years, measured as the difference between the 75th percentile and the 25th percentile, consistently vary across the months in the three uppermost soil layers and it reaches its maximum variations during summer and autumn seasons. On the contrary, it remains almost constant throughout the year for the deepest soil layer. In general, the magnitude of the spread and its variability seems to be well represented by the seasonal forecasting system. Finally, moving across different forecast lead times, a largest bias can be found as the lead time increases, whereas the spread in general remains constant.

The most important result from Figure 5 regards the smaller variability of soil moisture in the deepest layer, with respect to that of the surface layers. As expected, the dynamics of the deepest soil layer is slower than the three uppermost layers and this may be important for a seasonal forecasting system where slowly varying variable can be a source of predictability (Kumar et al., 2019). In the following, we will use ERA5-LAND as main products for comparing with forecasts. However, all the analysis have been done also for ERA5 reanalysis in order to confirm that results are not significantly affected by the choice of the reanalysis system.

To confirm such results and show the different dynamics of the soil layers across all the central Mediterranean region, Figure 6 provides the memory time scale, as evaluated from ERA5-LAND daily mean soil moisture. The memory time scale in the first soil layer (Figure 6a) is between 1 and 2 months, with minimum values of 5 days over complex orographic region (Alps and Dinaric Alps) where fast oscillations of soil moisture occur, and maximum values of 64 days over the other regions. In general, the memory time scale increases for all regions as we move toward the deepest soil layer, even though in some areas it remains also below 40 days. In the soil layer 2, the maximum values of memory time scale is around 3 months, with some regions in the Alps and the dinaric Alps still exhibiting values as low as 6 days. In soil layer 3, minimum values equal to two weeks are

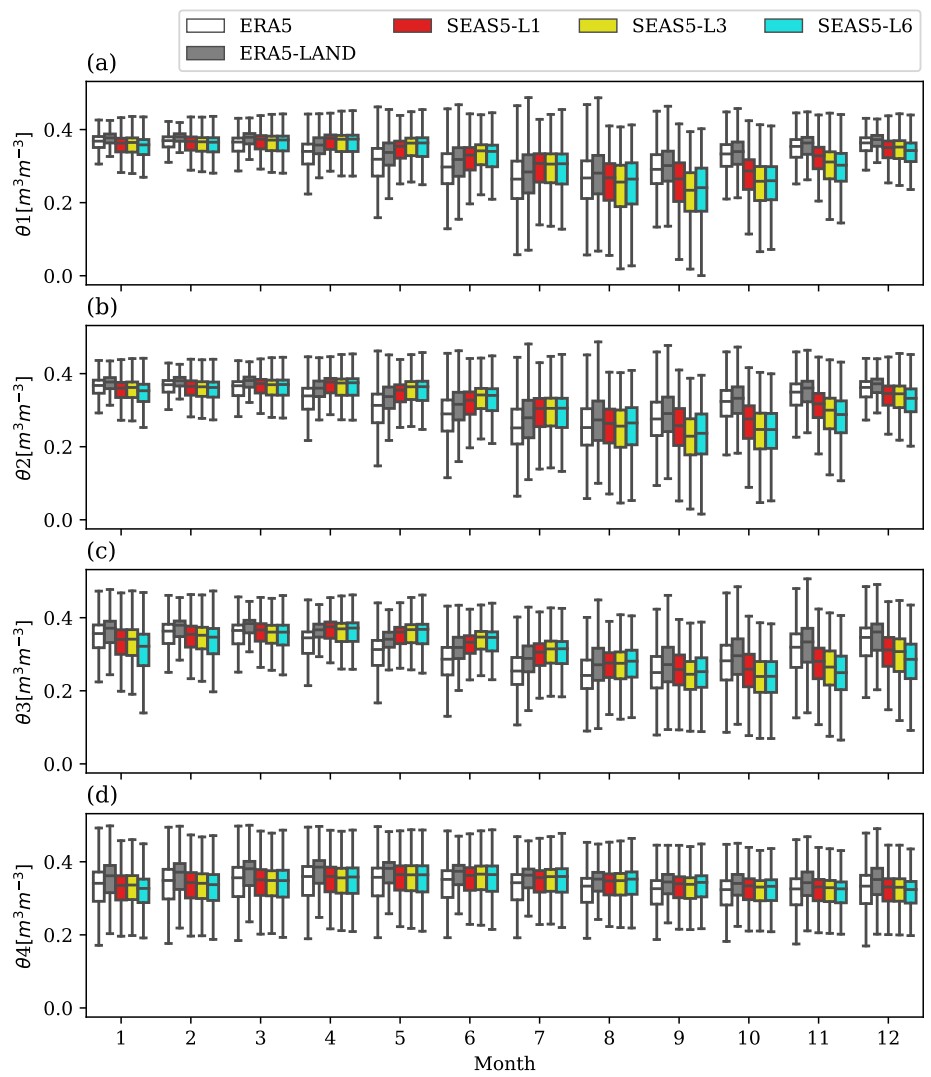

**Figure 5.** Boxplot of monthly mean soil moisture values for all the land grid points (land percentage > 75%) for ERA5 (white), ERA5-LAND (gray), SEAS5-L1 (lead time 1 month, red), SEAS5-L3 (lead time 3 months, yellow), SEAS5-L6 (lead time 6 months, cyan): (a) soil layer 1 (7 cm), (b) soil layer 2 (28 cm), (c) soil layer 3 (100 cm), and (d) soil layer 4 (289 cm).

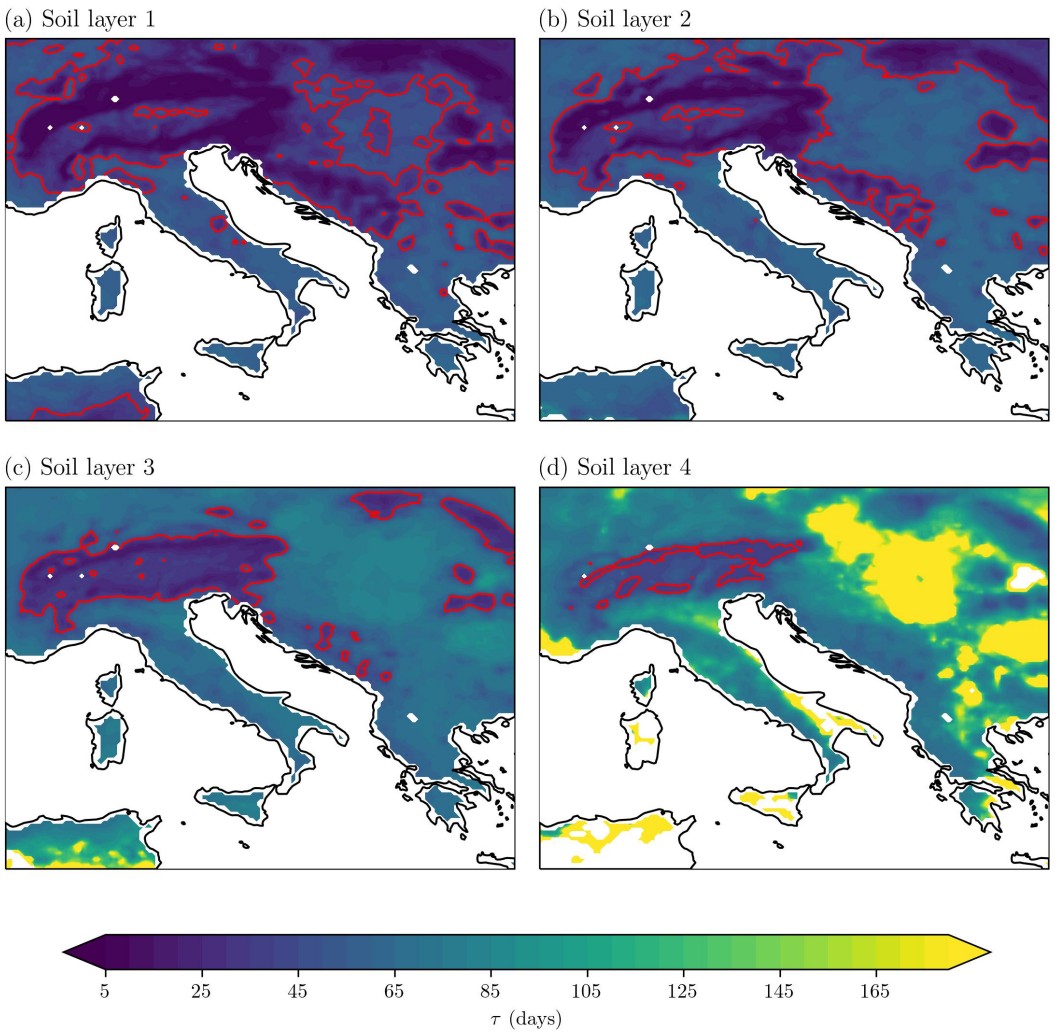

**Figure 6.** Memory time-scale of daily mean ERA5-LAND soil moisture over the period 2001-2020 for the Central Mediterranean for: (a) soil layer 1 (7 cm), (b) soil layer 2 (28 cm), (c) soil layer 3 (100 cm), and (d) soil layer 4 (289 cm). The time-scale is reported in days and corresponds to the time in which the temporal autocorrelation becomes lower than $e^{-1}$. Land regions where color shading is absent are regions where the memory time scale exceeds 1 year. Red contour line indicates where the memory time scale corresponds to 40 days.

still present in the Alps, while the maximum values even larger than 6 months appear in the Northern part of Africa. Finally, in the soil layer 4, the minimum time scale is 1 month in some region of the Alps, while the maximum values can exceed also the entire year (white areas in Figure 6d). There is a marked variability of soil moisture time scale in the fourth layer which exhibits a close connection to the orography of the domain (Figure 1b). Complex orographic areas, with the exception of few regions such the Northern Africa, usually exhibit smaller memory time scale than the flat areas.

## 4.2 Root-Mean-Squared Error (RMSE)

The RMSE of the seasonal forecasts ensemble mean SSMA over all soil layers is shown in Figure 7 for lead-times 1, 3 and 6 months. In all cases, the average magnitude error is almost larger than one standard deviation of soil moisture (1 SSMA) over soil layer 1 and soil layer 2 (Figures 7a and 7d). This error remains almost constant over different forecast lead-times.

Going towards the deepest soil layers and considering lead-time 1 month, the RMSE decreases over certain regions (Provence in France, Central and North Italy, Hungary and Romania), with values below 0.75, while it largely increases in other regions like the Alps, South-eastern Sicily, Sardinia and Tunisia (Figure 7l). The same distribution of average errors characterizes also lead-times 3 and 6 months, even if with a slight increase of RMSE over all regions. As a result, the accuracy of seasonal predictions increases for the deeper soil layers. This can be attributed to the slower dynamics of the deep soil layers as shown in Figure 6.

## 4.3 Anomaly Correlation Coefficient (ACC)

Figure 8 shows the ACC between forecasted and observed SSMA. As found for the RMSE, ACC reaches significant values (as indicated by black dots in Figure 8) above 0.8 (shaded contours in Figure 8) only on the deepest soil layer and over certain regions like Central and Northern Italy, some parts of France, Croatia and Hungary (Figure 8l). At lead-time 6 month, some regions like Central and Northern Italy and Bavaria, still exhibit high correlation values (Figure 8n). On the other hand, no correlation is found for the upper soil layers at 3 and 6-months lead-time (Figures 8b,c,e,f,h,i). This absence of correlation is also present at the deepest soil layer in the Alps, the Sardinia, the South-western coast of Italy, and in Tunisia, where correlation coefficient become negative (Figure 8n). At 6-month lead-times, the correlation disappears also for all the Western coast of Balkan peninsula (Figure 8n), where there are positive values at 1 and 3 lead-time month (Figure 8l and 8m, respectively).

Figure 9 shows the average ACC for all forecast months and lead-times. The first column shows values averaged over all domain, while the second column shows values averaged over Central Italy (black squared shown in Figure 1). Either averaging over all the domain or only over Central Italy, correlation is evident only in the deepest soil layer (SSMA4) is considered. With regard to the correlation of SSMA4 forecasts with observations, the domain-average ACC is always below 0.8, while it increases above 0.8 over Central Italy. In general, the highest correlations are found over the Autumn (SON) season, while the lowest are during the winter (DJF) season. In areas with a high correlation, like Central Italy, the most correlated target-months are between April and October, with a minimum in December and January.

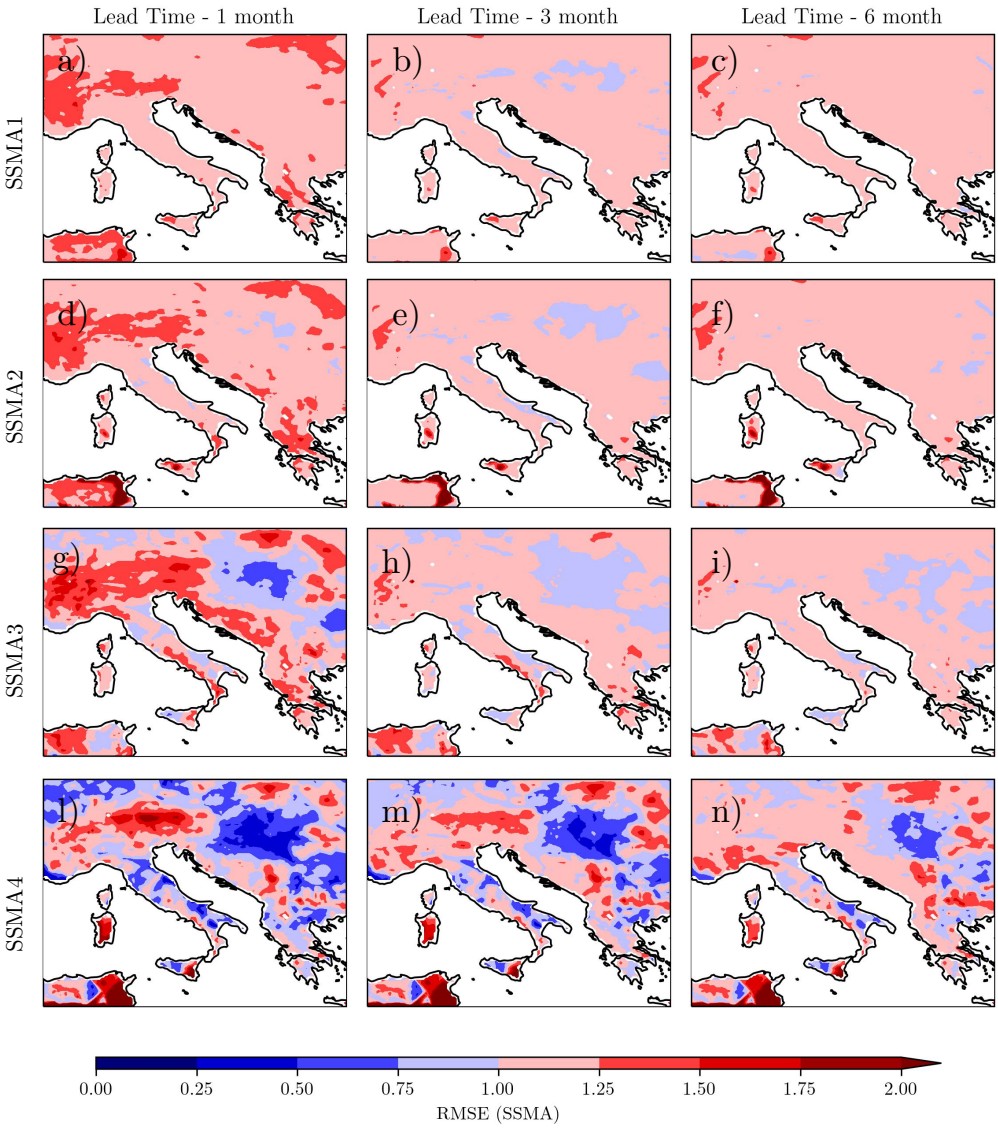

**Figure 7.** RMSE of Standardized Soil Moisture Anomalies (SSMA) averaged over the whole analyzed period (2001-2021). Rows show different soil layer: (a,b,c) soil layer 1 (7 cm); (d,e,f) soil layer 2 (28 cm); (g,h,i) soil layer 3 (100 cm); (l,m,n) soil layer 4 (289 cm). Columns show the same statistics for the forecast values at different forecast lead-times (1, 3, and 6 months).

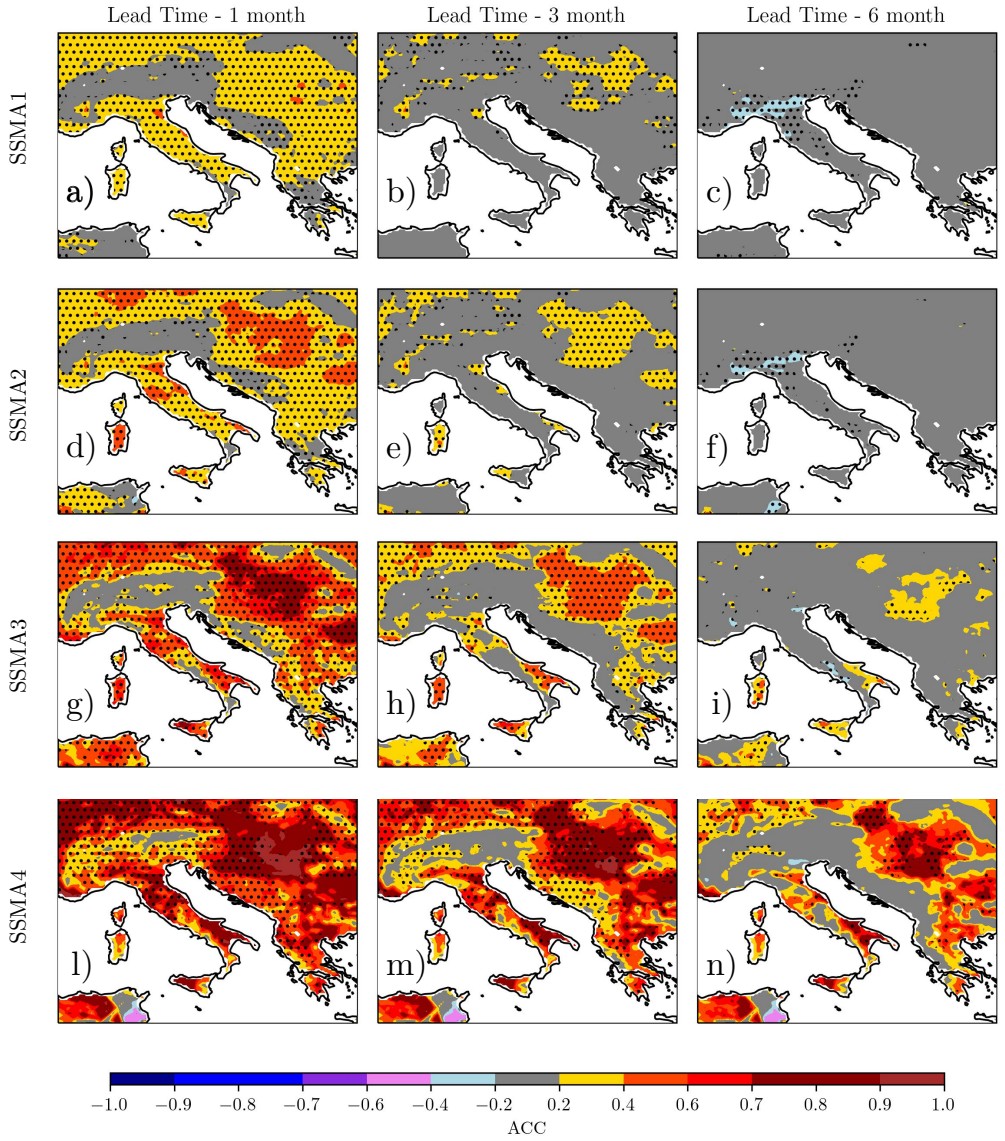

**Figure 8.** Anomaly Correlation Coefficient (ACC) of Standardized Soil Moisture Anomalies (SSMA) averaged over the whole analyzed period (2001-2021). Rows show different soil layer: (a,b,c) soil layer 1 (7 cm); (d,e,f) soil layer 2 (28 cm); (g,h,i) soil layer 3 (100 cm); (l,m,n) soil layer 4 (289 cm). Columns show the same statistics at different forecast lead-times (1, 3 and 6 months). Significant correlation (p-value < 0.05) are marked with black dots.

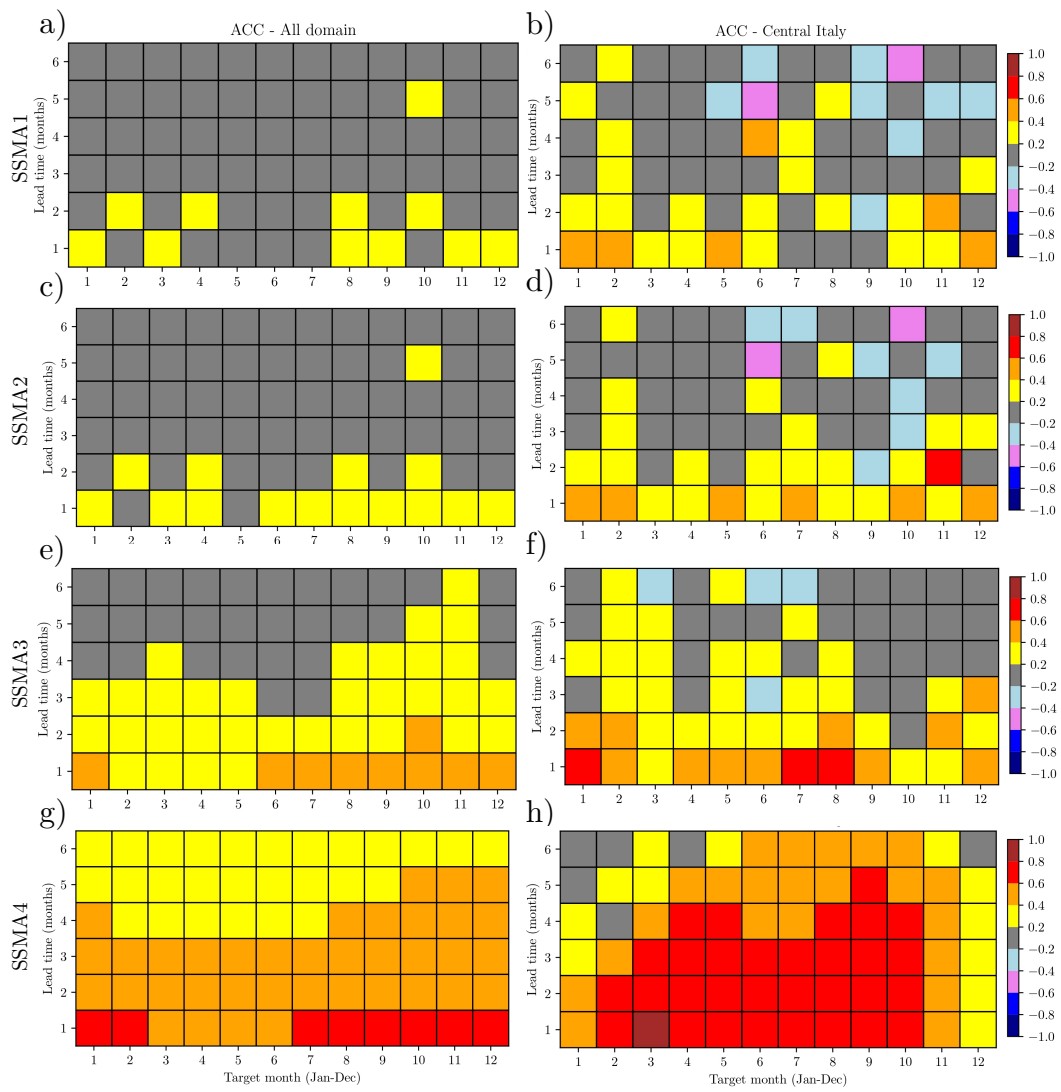

**Figure 9.** Area averaged Anomaly Correlation Coefficient (ACC) for each target month (x axis) and for different lead-times (y axis). Average values are computed over the whole domain (a,c,e,g) and Central Italy (b,d,f,h, black squared areas reported in Figure 1a). Rows show different soil layers: top layer (SSMA1, 7 cm depth), second layer (SSMA2, 28 cm depth), third layer (SSMA3, 100 cm depth), and bottom layer (SSMA4, 289 cm depth).

## 4.4 Relative Operating Characteristic (ROC)

The ability of the seasonal forecasts to discriminate between dry and wet events is examined trough the area under the ROC curve. In this paper, a wet and dry event is considered as the one in which SSMA is above or below 1, respectively. A ROC area larger than 0.5 means that forecasts can give more information than climatology alone, thereby indicating the potential usefulness of the forecasts. Figure 10 shows the ROC area for dry events for the first three soil layers at lead-time 1, 3, and 6 months, respectively. As also found for RMSE and ACC, it is evident that the forecast becomes more effective going towards the deepest soil layers. Values larger than 0.8 concern only SSMA2 and SSMA3 and some regions (i.e., Central and Northern Italy, internal areas of Hungary). The values in question exhibit a decline as the forecast lead time increases. This trend reaches its maximum at lead time six months, at which point no skill is evident in any region or soil layer. The sole exception to this is found in some regions of southern Europe, namely Northern Africa, Apulia and Sicily. The same behavior is observed for wet events, but with smaller values of ROC area, indicating that wet events are less predictable than dry events (not shown). From Figure 10 it is evident that seasonal forecast for the upper three layers is useful only in certain regions like the Central and Northern part of Italy and some internal area of Hungary, and only for shorter lead-time months. There are also some areas which exhibit no skill at all, neither at different layers or different lead-time months: the South-western coast of Italy, the Southern part of the Balkan peninsula, and the Alps.

The picture changes for the deepest soil layer 4 as shown in Figure 11. At lead-time 1 month, dry and wet periods show similar spatial distribution of ROC area, but with dry events (Figure 11a) having larger values than wet events (Figure 11b). Areas with no skill are still present and they are very similar to those listed above for the other soil layers: South-western coast of Italy, the Alps, Tunisia, the Alps and the Southern portion of Balkan Peninsula. There are also regions, like Sicily and Sardinia, where the ROC area is larger than 0.5 for dry events, but it turns into values smaller than 0.5 for wet events.

When examining lead-time 6 months, there are some areas where the seasonal forecasts are still very useful and exhibit large ROC area: Provence, South eastern coast of Italy, Central and Northern Italy, internal areas of Balkan peninsula. Instead, other areas loose their predictability, such as the Adriatic coast of Balkan peninsula or the Alps.

## 4.5 Sensitivity to soil moisture preconditions

In order to better understand the system's predictive ability to different soil moisture conditions, we study wether the forecast performance varies with varying antecedent soil moisture preconditions. In Section 4.1, we demonstrated that the memory time scale of soil moisture in the deepest soil layer is, on average, approximately three months. Consequently, we will consider the soil moisture antecedent condition to be that which was three months earlier. In order to have a larger number of events, differently from the previous section, we will consider dry events and wet events as those where the SSMA was lower than -0.5 and higher than 0.5, respectively. The dry precondition is considered to be present when the antecedent SSMA is negative, while a wet precondition is verified when the SSMA is positive. An example of event selection for the Umbria reference point is reported in Figure 12.

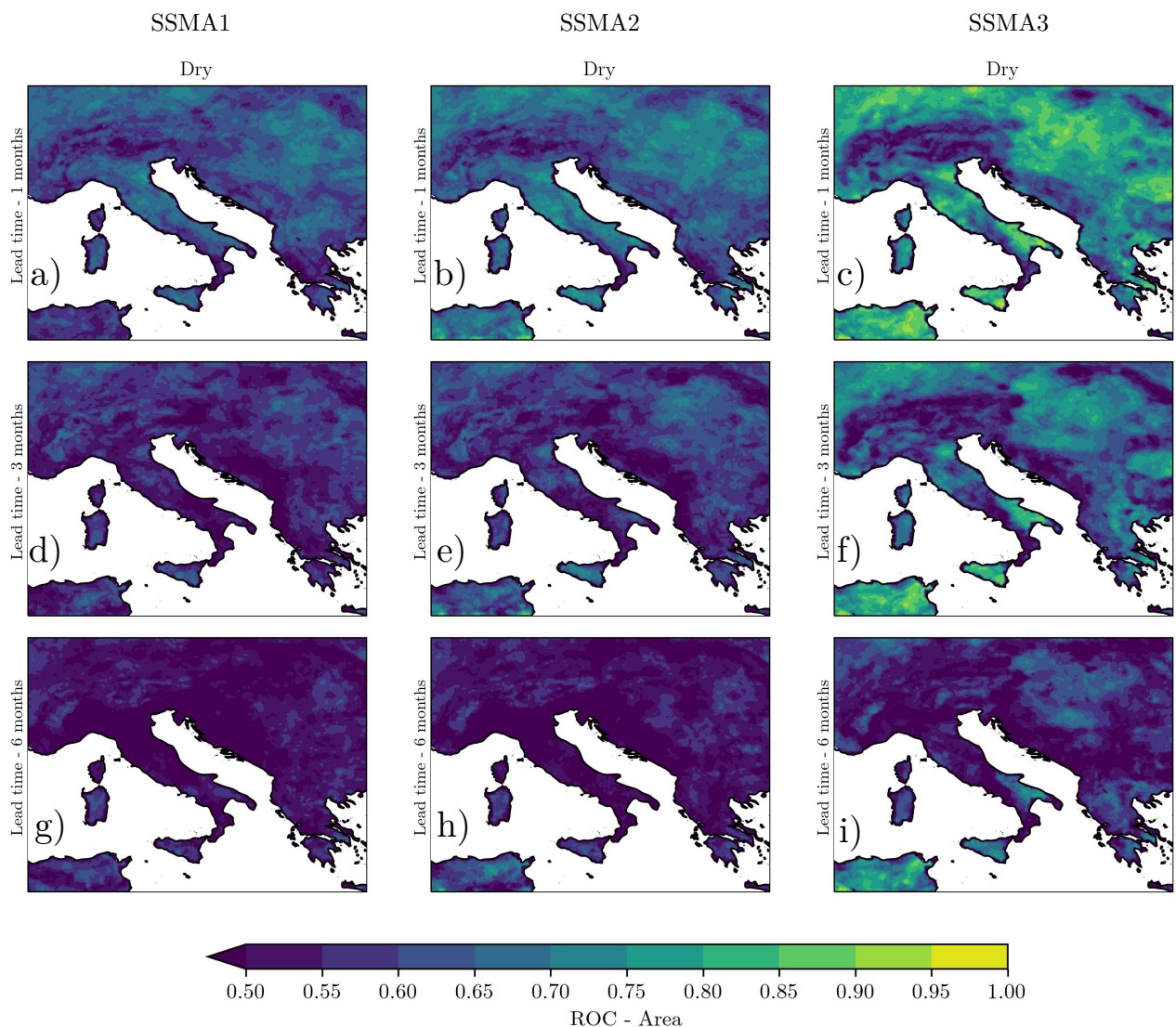

**Figure 10.** Area under the ROC curve averaged over all dry events during 2001-2021 for: (a,d,g) soil layer 1 (7 cm) SSMA1; (b,e,h) soil layer 2 (28 cm) SSMA2; (c,f,i) soil layer 3 (100 cm) SSMA3. Different rows concern different lead-times.

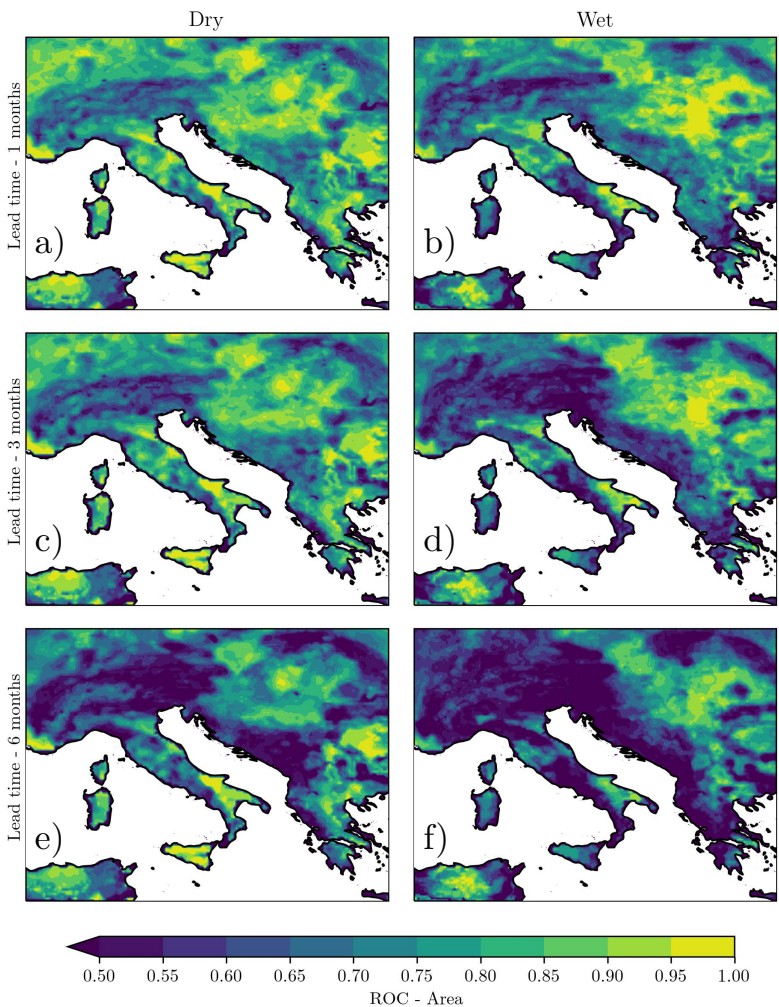

**Figure 11.** Area under the ROC curve averaged over all dry (a,c,d) and wet (b,d,g) events during 2001-2021 for the deepest soil layer 4 (289 cm) SSMA4; different rows concern different lead-times.

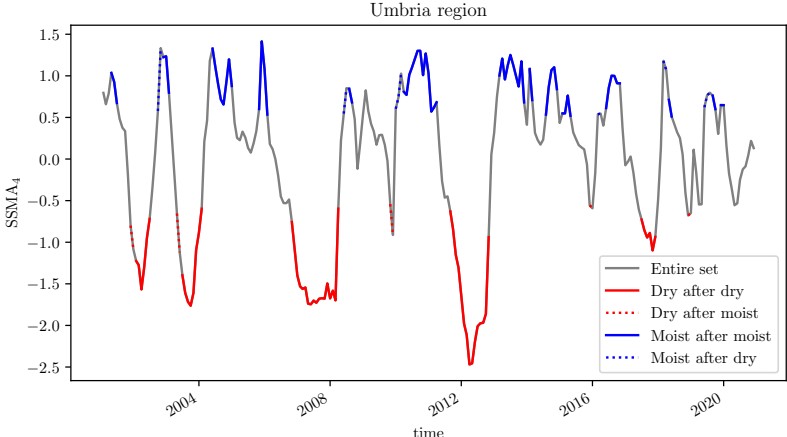

**Figure 12.** Example of event selection based on soil moisture pre-condition over the Umbria reference point for ERA5-LAND SSMA in the deepest soil layer 4 (289 cm). Dry periods are considered as those with SSMA $< -0.5$, while wet periods are those with SSMA $> 0.5$. A period is considered to happen after a dry period when the SSMA evaluated three months earlier is negative (SSMA $< 0$), while a period is considered to happen after a wet period when the SSMA evaluated three months earlier is positive (SSMA $> 0$). The result of such a selection is reported with the following lines: entire time series (gray solid line), dry period after a dry period (red solid line), dry period after a wet period (red dotted line), wet period after a dry period (blue dotted line), wet period after a wet period (blue solid line).

After the event selection, we calculate the area under the ROC curve for each grid point, following the procedure already used for dry and wet events in Section 4.4. The results are reported in Figure 13 for the deepest soil layer and considering only the forecast at lead time 1 month.

As expected, most of the predictive ability of the system comes from the memory of the deepest soil layer itself, since
the area under the ROC curve is larger on average for the events where antecedent condition is correspondent to the present condition (dry after dry, wet after wet). When the system is in transition from a dry period to a wet period, only few regions exhibit values of the area under the ROC curve larger than 0.7: some regions of central and northern Italy, internal regions of the Balkan peninsula and the Hungary region. In particular, the Hungary region (the Great Hungarian Plain) seems to have a large values of the score for all the events type. Regarding the wet after dry period the score is relevant also for the southern-east
coast of Italy and the Wallachia in Romania.

## 5   Links with groundwater levels: the 2012-2013 dry and wet periods

In this section, we shows a possible application of seasonal forecasts of soil moisture for groundwater management.

Figure 14a shows the water table level observations (expressed as a standardized anomalies with respect to their mean and standard deviation) in two different locations of Italy, Umbria and Veneto regions, in the Central and Northern part of Italy,
respectively (Figure 1). The monitored aquifers are selected to be shallow (depth smaller than 10 m) and unconfined in order to

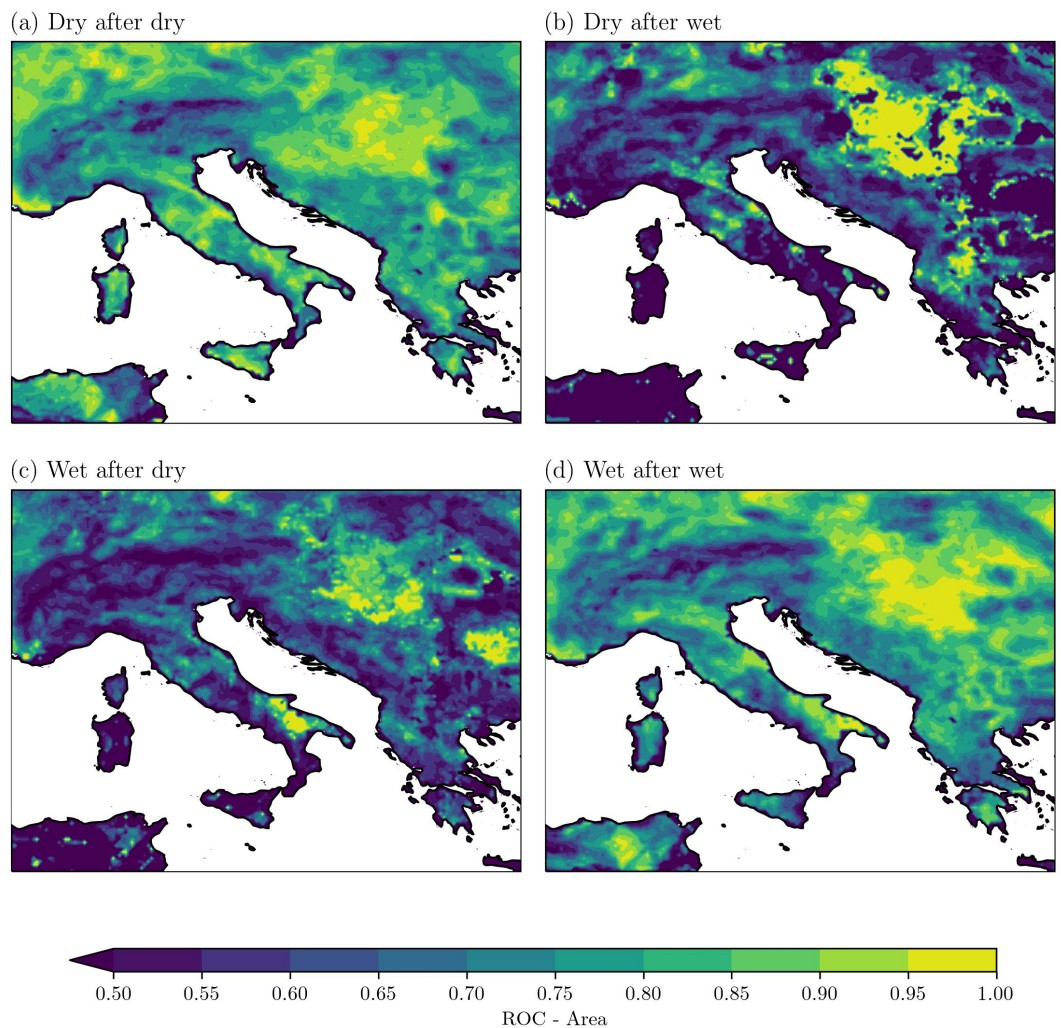

**Figure 13.** Area under the ROC curve for the deepest soil layer 4 (289 cm), SSMA4 and for the forecast lead time 1 month as averaged over all: (a) dry after dry events; (b) dry after wet events; (c) wet after dry events; (d) wet after wet events.

be directly influenced by atmospheric conditions rather than other groundwater processes (Bongioannini Cerlini et al., 2021). From such observations we detect only three dry periods (in 2007, 2012 and 2017) where the standardized anomalies of the water table level were less than -1 for both regions. On the other hand, wet periods in the water table observations, where values larger than 1 are observed, seem to happen more frequently.

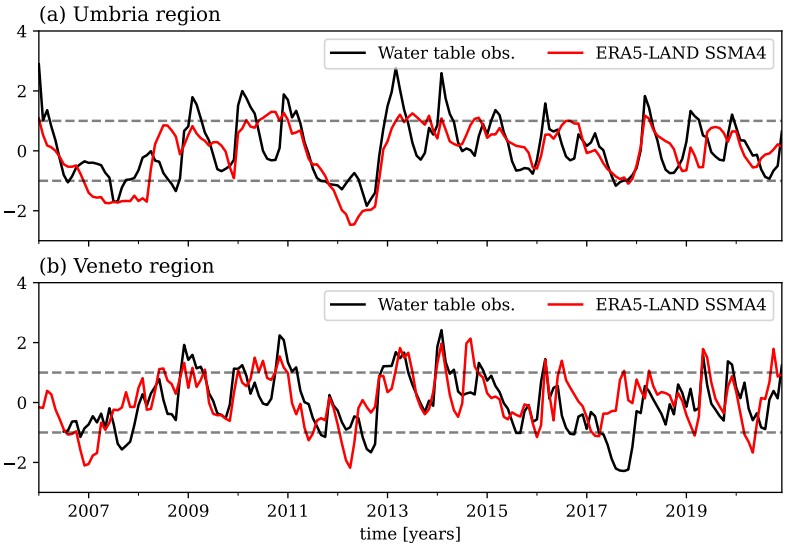

**Figure 14.** The correspondence between standardized anomalies of the water table elevations from a piezometric network and SSMA4 from ERA5-LAND reanalysis for two points of the Mediterranean region: (a) Umbria region and (b) Veneto region, as shown in Figure 1).

The water shortage in 2007, 2012 and 2017 in different regions of Italy is an indication of the synoptic scale character of such drought periods. The variability of water table level is well captured by the variability of deep soil moisture as extracted from ERA5 reanalysis, as shown in Figure 14b. In 2007, 2012 and 2017 negative anomalies are observed also for SSMA4 in both the analyzed piezometers, with the 2017 anomaly being weaker with respect to the others.

In the below analysis, we focus on the 2012 dry period and the following wet period in order to test the ability of seasonal forecasts to predict such events. Figures 15a, 15b and 15c show the spatial distribution of SSMA4 over the Central Mediterranean on June 2012, December 2012, and June 2013, respectively. These periods are taken as a reference for the start of the dry period, the end of the dry period and the start of the wet period, as observed in Figure 14b for North-Central Italy.

June 2012 is characterized by a large negative anomaly over all the domain expect for the Alps, Sicily, Tunisia and the Adriatic coast of the Balkan Peninsula. Seasonal forecasts for lead-time 1 month predict smaller negative anomalies over Central Italy and the Balkan peninsula, while largely underestimating the positive anomalies over the Alps (Figure 15d). The forecast slightly improves in Northern Italy and Balkan peninsula going to lead-time 3 and 6 months (Figure 15g and 15l), while it gets

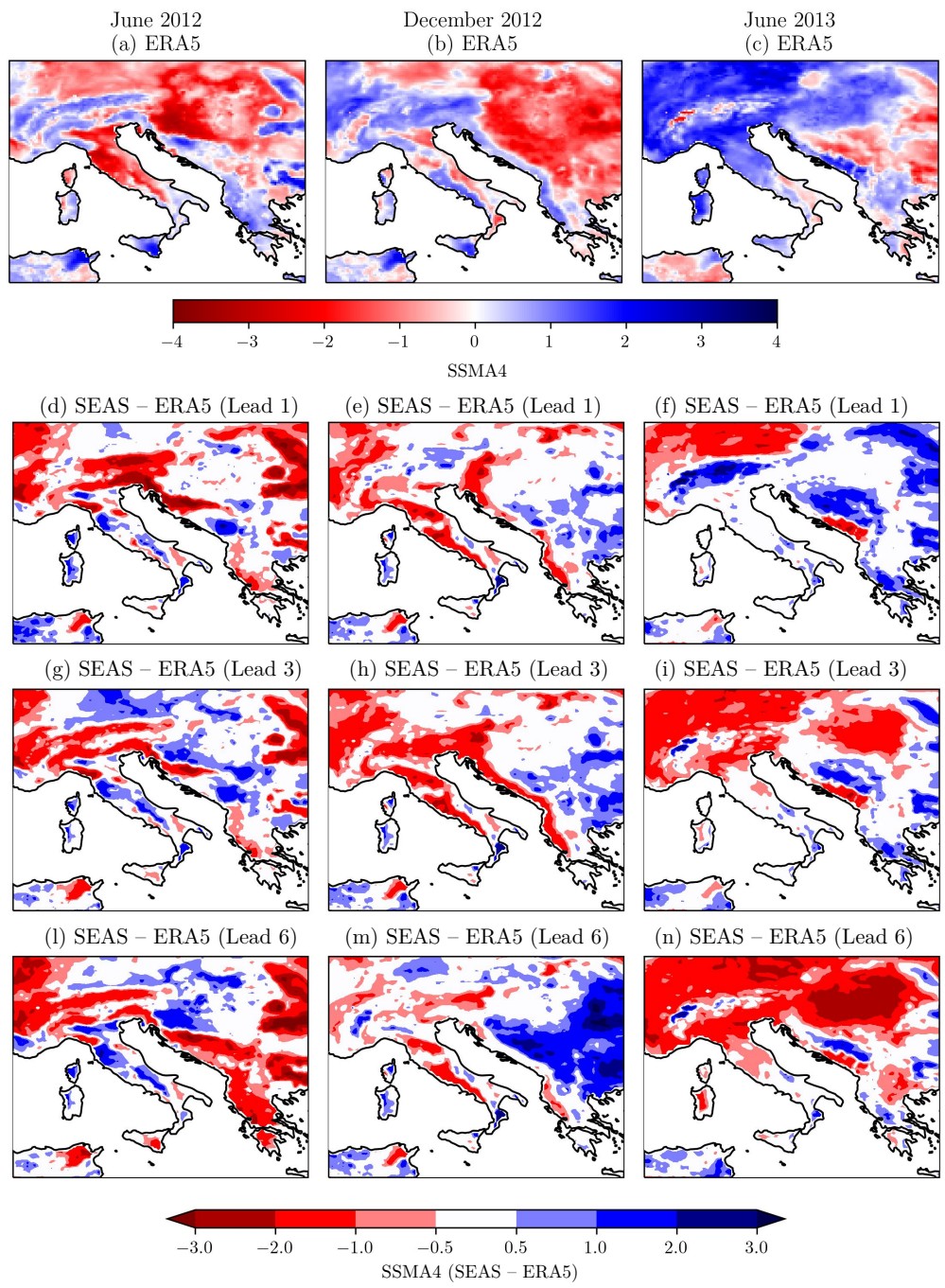

**Figure 15.** Spatial distribution of observed ERA5-LAND (a,b,c) and forecasted SSMA4 anomalies (d-n) during the analyzed case studies: first column for the dry period of June 2012, second column for the dry period of December 2012, and third column for the wet period of June 2013. Figures (d,e,f) concern forecast lead-time 1 month, (g,h,i) lead-time 3 months and (l,m,n) lead-time 6 month.

worse for Sicily and Tunisia.

December 2012 shows a similar spatial distribution of SSMA4 except for larger positive anomalies on the Alps, the South-western coast of Italy, and the South-western coast of the Balkan Peninsula (Figure 15b). Also the amplitude of negative anomalies of SSMA4 decreases in Central and Northern Italy. The seasonal forecasts perform well in Central and North Italy, in the South-eastern coast of Italy, in Sicily and in Provence for all lead-times (Figure 15e-h-m). However, it was not able to detect the large increase in positive anomalies over the Alps, the Western coast of Balkan peninsula, and the Tyrrhenian coast

of Italy. Also the larger negative SSMA4 in the internal regions of the Balkan peninsula was not detected.

The wet period of June 2013 involved especially the Northern part of the domain with large positive anomalies of SSMA4 (Figure 15c). Seasonal forecasts show in general a good performance especially in Central and North Italy at lead-time 1-month (Figure 15f), while they tend to underestimate such positive anomaly at larger lead-times, especially over Hungary (Figure 15i-n).

## 6 Discussion and conclusions

This study provides a first assessment of seasonal forecast of soil moisture for the Central Mediterranean Region. The seasonal model analysed in this study is SEAS5, whereas the reanalysis is ERA5-LAND, both produced by the ECMWF. ERA5-LAND is considered as a reference dataset for soil moisture observations. Twenty-five member seasonal forecasts with lead-times from

400 1 to 6 months have been analyzed from 2001 to 2021, by considering the hindcast period 2001-2016 as climatology. In this reference period, the standardized soil moisture anomaly (SSMA) has been evaluated and the forecasts have been bias-adjusted trough the mean-variance adjustment method. Then Root Mean Squared Error (RMSE), the Anomaly Correlation Coefficient (ACC) have been evaluated for SSMA for all soil layers considered in ERA5-LAND. To test the ability of the forecast to discriminate between dry and wet events, the Relative Operating Characteristic (ROC) area has been calculated. Finally, a case

study of the dry and wet periods during 2012-2013 has been studied in detail, to show the potential usefulness of the seasonal model. The outcomes of the research can be summarized as it follows.

As indicated by the RMSE, the average magnitude of the forecast errors decreases as we go deeper into the soil. Only in the deepest soil layer at 289 cm depth, the RMSE can reach values below 0.5 even for lead-time 6 months. However, this is

410 valid only over certain regions like Central and Northern Italy, Hungary, some internal regions of the Balkan Peninsula and the Provence region. The RMSE remains too large in other regions, even when considering only the deepest layer. Significant values of the ACC, with values larger than 0.8, can be found over the mentioned regions even at lead-time of 6 months. The analyzed performance depends on the memory time scale of the soil layer: the higher the memory time scale the higher the forecast performance. The main physical factors affecting the spatial variability of memory time scale are various: soil depth,

orographic complexity, local climatology (e.g. soil aridity, mean precipitation) and soil hydraulic properties. In general, we found better forecast performance in the deepest soil layer and in regions with low orographic complexity, corresponding to

regions with larger memory time scale. This is in agreement with previous studies on the spatial variability of soil moisture memory. For example, MacLeod et al. (2016) found a large sensitivity of soil moisture memory on soil hydraulic parameters (e.g. Brunone et al., 2003) and found a longer memory in the deepest soil layers. The dependence on soil depth was ascribed to the smallest influence of the throughfall precipitation, which is partly absorbed by evapotranspiration before penetrating into the deepest soil layers. Moreover, Orth et al. (2013) analyzed the influence of altitude, topography and dryness index on soil moisture memory time scales, finding that memory time scales decrease with elevation and increase with topography (measured by a topographic index which is a function of the slope) and aridity. Our study identified comparable signals in the Central Mediterranean. However, further investigations are required to ascertain which factors (soil properties, altitude, orographic complexity, climate) are most influential in shaping the soil moisture memory in a given region. Such an investigation could inform a more robust modeling approach, incorporating additional parameter uncertainty into the forecasting system, which may ultimately enhance the skill of seasonal forecasts (MacLeod et al., 2016).

The ability of seasonal forecasts to detect wet and dry events exhibits a large variability within the domain. However, a ROC larger than 0.8 can be found in certain regions for the deepest soil layer also for lead-time 6 months. This indicates that in those regions, like Provence, Central and North Italy, the South-eastern coast of Italy and the internal regions of Balkan peninsula, seasonal forecasts can be used to detect such events in advance. The area under the ROC curve for dry and wet events in the two uppermost soil layers is about 0.5 when lead times exceeding three months are considered. This suggests that the seasonal forecasting is not a reliable method for predicting the evolution of upper soil moisture beyond three months. A small ROC area for dry and wet events is found at lead-time 6 months especially in mountainous regions (Alps and Dinaric Alps), confirming the spatial variability already found for RMSE and ACC indicators. In general, for all soil layers, dry events are generally better captured than wet events. From the water management point of view, this indicates that information provided by seasonal forecast on soil moisture should be trusted more for supporting drought-risk management rather than flood-risk management. The most useful forecasts are produced for events where the antecedent and present conditions are aligned (e.g., dry after dry, wet after wet). This further validates the significance of soil moisture memory and soil moisture pre-condition for the predictability of the system.

In the areas with large correlation coefficients, the larger correlations are found between April and October, while a minimum correlation is found in December and January. Such a feature is of great relevance in terms of water resources management as the critical period is late spring and summer, when the water demand is the largest in the year for both agriculture and civil activities. As an example, the case study of 2012 drought period shows how the SEAS5 model is able to predict such an event for Central and Northern Italy 6 months before. Moreover, the strict connection between the deepest soil moisture and the water table of shallow unconfined aquifer in Italy, highlights the large potential usefulness of seasonal forecasts of soil moisture for water management purposes.

A local water management service, especially those located in the most effective areas, could monitor the forecasted soil moisture anomalies across all soil layers, as publicly provided by the Copernicus Climate Data Store. The seasonal forecasting system can provide a probability of either a wet, dry, or normal month at different lead times, thus assisting in the decision-

making process for the management of drought or flood risks. Moreover, groundwater models or simpler methods such as those in Bongioannini Cerlini et al. (2021) could be run starting from forecasted soil moisture products for monitoring groundwater levels in unconfined aquifers. In this case, when using these data in very local application, an evaluation of the influence of irrigation input could be of great importance. However, the volume of water used for irrigation – a critical quantity towards water resources management – is a data that is very difficult to find for a number of reasons. One of the most important is the poor measuring instrumentation installed in irrigation systems. Consequently, it is difficult to estimate the contribution to soil moisture. The irrigation volume being equal, the dispersion towards the aquifer depends on the type of irrigation practiced. Maximum dispersion occurs in flowing systems, whereas in the case of drop irrigation in pressurised networks, dispersion can be considered negligible. For the Veneto irrigation systems, due to the extreme relevance of aquifers, reliable quantitative assessments are available (Altissimo et al., 1999; Rinaldo et al., 2010). These irrigation systems, active for the entire year, are supplied by surface water and are of the flowing type with unlined channels in 70% of the cases. Groundwater withdrawals are carried out by water utilities for drinking water use. On the basis of the data provided by the land reclamation consortia and water utilities, it is shown that the entity of dispersion of the irrigation systems, minus withdrawals for drinking water use, is comparable to that of effective rainfall (Altissimo et al., 1999; Rinaldo et al., 2010). In the Umbria region, even if analogous documentation is not available, the same situation can be assumed. However, since in this study the analysis is focused on the soil moisture anomaly, the contribute of the irrigation volume is not effective, as it is almost constant over the year. On the contrary, in cases where irrigation were active only in few months, its effect should be taken into account provided that data availability allows.

To refine the proposed approach two possible paths can be followed in future research. The first is to use different seasonal forecasting models, different reanalysis and observation products (e.g. MERRA-2 (Gelaro et al., 2017) and GLEAMv3 (Martens et al., 2017)). The second path is to analyze in more detail the behavior of the autocorrelation function of soil moisture anomalies across different soil layers. This to evaluate the role of the seasonal cycle and the reemergence of soil moisture as hypothesized by Kumar et al. (2019). Then, this would allow us to compare the seasonal forecast performance with those obtained by memory-prediction models, following the approach proposed by Esit et al. (2021).

*Code availability.* The python packages xarray and xskillscore have been used extensively in this work and they are freely available at https://docs.xarray.dev/en/stable/ and https://xskillscore.readthedocs.io/en/stable/

*Data availability.* ERA5 and ERA5-LAND reanalysis data and Seasonal forecasts data are available on the Copernicus Climate Data Store. The water table data of Umbria that support the findings of this study are available upon request from https://apps.arpa.umbria.it/acqua/contenuto/Livelli-Di-Falda.

*Author contributions.* All authors contributed to the conceptualization of the research. L.S. carried out the data analysis. All authors contributed to the investigation of the results. L.S. wrote the original draft. P.B.C. supervised all the research group work. All authors reviewed and edited the manuscript.

*Competing interests.* The authors declare that they have no conflict of interest

*Acknowledgements.* This research has been supported by *Fondi di Ricerca di Ateneo - edizioni 2021 e 2022* of the University of Perugia and SSTAM. P.B.C. has been funded by the European Union - NextGenerationEU under the Italian Ministry of University and Research (MUR) National Innovation Ecosystem grant ECS00000041 - VITALITY (CUP J97G22000170005). The Authors would like to thank the European Commission, MUR (Italy), Fapesc (Brazil), and FCT (Portugal) for funding in the frame of the collaborative international consortium MORE4WATER financed under the 2022 Joint call of the European Partnership 101060874 – Water4all (CUP J93C23002030006). The
authors also acknowledge Marco Sangati and Sinergeo Srl for providing data of the water table level in Veneto region. L.S. was supported by the Italian Ministry of University and Research (MUR), through the PRIN 2022 PNRR Project P20229KW2R - SEAPLANE - "Simulation and modelling of interface fluxes in wind-wave flows for an improved climate science", CUP E53D23017010001, funded by the National Recovery and Resilience Plan (PNRR), Italy, Mission 04 Component 2 Investment 1.1 – NextGenerationEU.

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
