# Peer review of "Assessment of seasonal soil moisture forecasts over Central Mediterranean"

_EGUsphere, 2024_

## Author Response (AR1)

**EDITOR COMMENT**

Dear Authors,
Your original submission was subjected to a peer-review process by three experts, who provided valuable feedback on your study and facilitated productive discussions during the initial stages of the journal's evaluation. Two of the three reviewers provided positive feedback on your study, whereas Ref.#3 offered a more critical assessment and recommended rejection. Specifically, Ref. #3 raised concerns regarding the scientific rigor of the study. In my estimation, the criticisms leveled by Ref. #3 were more a consequence of a lack of explanations regarding the observed outcomes of the study than a weakness of your analyses. For instance, it would be beneficial for the authors to provide the readers with more in-depth explanations of the greater soil moisture values observed in the deeper soil layers. This outcome may be an artifact of the study, or alternatively, it may reflect an actual situation, largely since, on average, and apart from the soil physical-chemical properties, the uppermost soil layers are subjected more to evapotranspiration fluxes. Does the term "soil" still apply everywhere at a depth of 289 cm? It is similarly important to conduct an accurate evaluation and interpretation of the irrigation issue (where and how much?), with a particular focus on groundwater recharge. In conclusion, it is recommended that the authors provide a detailed rationale for their findings, taking into account the specific environmental conditions of the local areas.
The submission is released under major revisions and the authors are required to upload detailed point-by-point replies to all the reviewers' comments received thus far. Furthermore, in their responses, the authors are also encouraged to clarify those points that require more detailed interpretations of the presented results.

Dear Editor,

In the revised version of the paper we addressed your suggestions and the referee requests as indicated in the below replies. In particular,

1) We improved the reference dataset, by considering ERA5-LAND instead of ERA5 reanalysis, since it provides a better representation of the soil moisture (as suggested by Reviewer 2);

2) We discuss the obtained results in terms of the memory time-scale and the soil moisture variability of each soil layer over the whole domain (as suggested by Reviewer 2 and 3). Therefore we add an additional section to the Results (section 4.1);

3) We analyzed the dependence of forecast performance on the antecedent soil moisture content condition. Therefore we add another section to the Results (section 4.5)

4) We discuss the potential implications of Irrigation for soil moisture and groundwater level changes and we add a complete discussion of the results (Section 6)

According to the above changes, we decided to include two more authors (Bruno Brunone and Silvia Menicone) that were needed to address the rewiewers referee requests. Moreover we modify the title from "Assessment of seasonal soil moisture forecasts over Central Mediterranean toward groundwater management" into "Assessment of seasonal soil moisture forecasts over Central Mediterranean" to reflect more properly the content of the paper.

To pointed out the changes in the revised version of the paper, a tracked-changes version is also included.

In the following we summarize the main changes in the revised version of the paper, according to the changes you requested (see also the tracked-changes version).

**CC1: Giacomo Medici**

General comments

Good research in the field of surface/groundwater interaction with angle on variations of the climate. Some detail is missing, please integrate my specific points.

We thank Dr. Giacomo Medici for his useful comments.

Specific comments

Lines 18-95. Any link between your research and transient groundwater flow models? I think so looking at the results Figure 9. Please, specify this point.

Most Land Surface Models, as HTESSEL that is used by ERA5 reanalysis, extend from the surface to a soil depth between 2 and 4 meters, constraining the flux at the bottom of the soil domain by applying a free-drainage condition, where the flow is only controlled by gravity. Our method, by using the strong correlation between such predicted bottom fluxes and the observed water table level [1] in shallow unconfined aquifers, avoid the use of additional transient groundwater model that are fully coupled with the land surface. Altough such groundwater models fully coupled with land surface models are found to generally improve the prediction of surface water fluxes (runoff and evapotranspiration) [2], they will also increase the computational cost. Therefore our method offer an easy-to-implement (where water table observations are already available) and low-cost alternative to the fully-coupling modeling option.

[1] Cerlini, P. B., L. Silvestri, S. Meniconi, and B. Brunone, 2021: Simulation of the Water Table Elevation in Shallow Unconfined Aquifers by means of the ERA5 Soil Moisture Dataset: The Umbria Region Case Study. Earth Interact., 25, 15–32, https://doi.org/10.1175/EI-D-20-0011.1

[2] Batelis S-C, Rahman M, Kollet S, Woods R, Rosolem R. Towards the representation of groundwater in the Joint UK Land Environment Simulator. Hydrological Processes. 2020; 34: 2843–2863. https://doi.org/10.1002/hyp.13767

Lines 59-60. "There is high confidence that the Mediterranean region will suffer from an increased aridity and an increase in hydrological droughts". Please, specify that aridity can heavily impact the snowmelt recharge of the aquifers in the mountain ranges of the Mediterranean area. Insert relevant literature on this point:

- Lorenzi, V., Banzato, F., Barberio, M. D., Goeppert, N., Goldscheider, N., Gori, F., Lacchini, A., Manetta, M., Medici, G., Petitta, M. (2024). Tracking flowpaths in a complex karst system through tracer test and hydrogeochemical monitoring: Implications for groundwater protection (Gran Sasso, Italy). Heliyon, 10(2).

- Doummar, J., Kassem, A. H., & Gurdak, J. J. (2018). Impact of historic and future climate on spring recharge and discharge based on an integrated numerical modelling approach: Application on a snow-governed semi-arid karst catchment area. Journal of Hydrology, 565, 636-649.

We will insert such comments and relevant literature in the introduction.

Line 95. Clearly state the specific objectives of your research by using numbers (e.g., i, ii and iii).

We will modify this.

Line 153-154. "Mean depth of water table below 10 m". Unclear, please revise.

The mean depth is calculated as the average of observed water table level over all the available observation period. We will clarify it in the text.

Line 303. Possible to disclose the areas of large correlation coefficient?

We will specify those areas: Provence, South eastern coast of Italy, Central and Northern Italy, internal areas of Balkan peninsula.

Lines 333-430. Please, integrate relevant literature on surface/groundwater interaction with links on climate variations in the Mediterranean region.

We will integrate the literature as suggested

Figures and tables

Figure 2. Difference in colour between the two types of green difficult to see. Possible to improve?

We agree with the referee, we will improve the colormap.

Figures 2 and 6, 7. All these maps should be larger.

Figure 7. What about the use of a dashed line?

Figure 9. Is it clear why the red lines are not continuous?

Figure 9. Insert reference to Figure 1 for the location of the 3 sites.

We will take care of all the above comments regarding the Figures' formatting.

**RC1: Referee #1**

In their article **"Assessment of seasonal soil moisture forecasts over Central Mediterranean toward groundwater management"**, the authors investigate the performance of SEAS5 soil moisture predictions across various lead times compared to ERA5 reanalysis, employing several performance metrics. The evaluation spans the period 2001-2021, focusing on the Central Mediterranean region. The findings reveal promising forecast accuracy for specific regions and soil layers, particularly at a depth of 289 cm.

**General comments**

Overall, the manuscript is well structured and well written. The topic of this research – the analysis of seasonal forecasts - is an area of relevance and interest. However, I have a couple of general comments: (1) There is a need for more detailed explanations and discussions on the performance metrics utilized in the study. Referencing other studies that employ similar metrics

would enhance the overall quality of the article. (2) To fully support the conclusions of this study, more case studies are needed. In their case study, the authors did not account for the antecedent moisture content, which is an important factor in understanding the dynamics of the studied phenomenon. I would greatly benefit the overall quality of this article to analyse additional events, e.g. to include case studies that examine wet/dry events occurring after a dry season versus after a wet season, to provide a more comprehensive analysis of the system's predictive ability to varying moisture conditions. Alternatively, comparing SEAS5 to additional soil moisture products such as SMAP, ESA-CCI, or in situ observations if available, would also support the conclusions of this research.

Dear referee,

we thank you for your efforts in reviewing our manuscript. In the following we will summarize how we will change the paper according to your request.

1) We included new literature regarding SEAS5 performance (in the introduction), as we did for ERA5. **Lines 68-82**

2) We included detailed explanations and literature on the performance metrics utilized in our study. Moreover we will plot the ROC curve in different point of the studied domain. **Section methods and Figure 4**

3) We are studying in detail additional periods that we will include as case studies in the revised paper. **See new section on moisture precondition Section 4.5**

4) After an examination of local sources of observations, we decided to not include other observational products in our study. The main reason is because the focus of the paper is on the deep soil moisture, while satellite products can observe up to 5 cm from the soil surface. Moreover in the considered study domain, there are few soil moisture observations from the International Soil Moisture Network, which are often maintained for short periods and they measure soil moisture only up to 45 cm. Therefore, regarding the evaluation of soil moisture below 5 cm depth and for an extensive period of 20 years, ERA5 reanalysis and water table observations are at the moment our best sources of observations.

**Specific comments:**

**Introduction:**

- This study is focussing particularly on SEAS5 performance. However, while ERA5 performance is stated and cited in the intro, the same is missing for SEAS5. Where do these forecast generally show best performance on a global scale? What are previous studies etc.

We will include new literature about SEAS5 performance in the new introduction

- What other climate services are available in general and specifically for this region? Are there already services available that are used by government/agriculture?

To the authors' knowledge, there are no climate services other than Copernicus, that provide seasonal forecasts of soil moisture for water resources management.

**Methods:**

- The metrics used here should be supported by more references and additional statements on their applicability, and interpretability.

This part of methods will be improved with more references and explanations.

- Why use the ROC metric and not plot the curve once?

We decide to focus in the spatial pattern of the ROC metric. However we will include the ROC curve in the revised paper.

- Are there no in situ observations of soil moisture available at all?

Please refer to point 4) in the answers to general comments.

**Results:**

- Overall, the results section is well structured. Some statements belong to the discussion or conclusion.
- It would be interesting to plot the performance of the whole ensemble of forecasts in one plot and comment on the spread.

We ask the referee for a clarification of the above comment. Is it about a specific figure/lines in the text? We will appreciate any specific suggestions or ideas on where to modify the results.

**Conclusion:**

- What is the relevance of this study, like development of real-time application for climate services as mentioned in Line 183 – 184. This is clearly stated in the title of but is not clear from the text. Mention again the relevance and goal of this study – drought risk, development for climate services. This is missing here.

We agree with the referee. We will modify the conclusion accordingly.

- In addition, the conclusion is missing an outlook. What is still missing for the development of climate services? What is the applicability of this study going forward? What is the applicability of this study going forward?
- Last comment needs more elaboration. This section should not be a summary of results but really dive into the limitations and outlook etc.

We agree with the referee. The conclusion will be more elaborated in the revised paper.

More specific and editorial comments are given below:

We thank the reviewer for all his comments. We will take care of all the specific comments below in the revised paper.

Line 8-10: Is this conclusion really supported by the results at hand? I suggest to soften or to analyse a larger variety of individual events to support this statement.

Line 15: Improvement in observations? Unclear! Do you mean in the availability of observations or do you mean in the quality of available data such as reanalysis (as you call reanalysis observations)? Please rephrase. This statement can be short in the abstract but needs to be supported by more elaborations in the conclusion. (see comments above on conclusion)

Line 21: I suggest reformulating "soil surface" to terrestrial surface or similar.

Line 23: I suggest removing "at the surface" form the sentence

Line 23 – 26: Unclear sentence structure, I suggest reformulating.

Line 27 – 30: Sentence unclear, please reformulate.

Line 32 – 34: Remove "the" after drive: "drives 90% of the inter-annual variability". Add more reference studies or soften the statement, e.g. "the variability of soil moisture simulations has been found to drive (…)"

Line 38 – 41: Shorten the sentence and use same tenses throughout the text. Suggestion: "In addition, Li et al. (2021) evaluated groundwater recharge estimations from different land surface models and found that the seasonal cycle of simulated groundwater storage (…)."

Line 47: Again, check for consistent usage of tenses. Which reanalysis products were compared?

Line 50 – 51: Please reformulate "regards (...)". For which product/land surface model?

Line 55: Which land surface model? I suggest removing this part (Line 54 – 57) as it is not really relevant to this study and makes the transition to the next paragraph a bit confusing/abrupt. (Alternatively, more elaborations are needed here to make the transition to next paragraph more comprehensive.)

Line 63: I suggest reformulating to something like: "(…) soil moisture is one of the most impactful land parameters and is crucial for the forecast skill. "

Line 68: I suggest reformulating: "This can be attributed to reduced variability (..). "

Line 75: This statement is false! Boas et al. looked at soil moisture predicted with LSM that was forced with atmospheric fields of SEAS. Please rephrase.

Line 87: "(…) to wet and dry events. "

Line 90: Accordingly, the paper is structures as follows. I suggest to shorten this whole paragraph substantially. This is not needed in this detail for a scientific manuscript.

Line 105: "Second, the complex orography of this region (…). "

Figure 1: Not the best choice of colormap. Please consider using a colormap that conforms to color blind standards.

Line 130 onwards: Why not use the bias adjusted version of ERA5?

Following suggestions from Referee #3 we will use also ERA5-LAND in the revised paper in order to validate the seasonal forecasts. Moreover we cannot find any bias adjusted version of ERA5 in the climate data store which contains the soil moisture at all soil levels (only surface soil moisture is usually available).

Line 137: Add reference citation for this statement.

Line 137 – 139: This belongs to methods section (is already mentioned there as well).

Line 152 – 153: "(…) with a mean water table depth below 10 m, (…)." Add reference citation.

Line 170 – 171, Equation is incomplete.

Line 183 – 184: Good statement but should be mentioned first in the introduction!

Line 200: I suggest reformulating this throughout the text: "1-month lead time"

Figure 2: Remove one "at " and missing parenthesis in Figure caption: Columns show the same statistics for the forecast values at different forecast lead times (1, 3 and 6 months).

Figure 4: "(..) over the whole domain (…)." I suggest adding the soil layer depth for all layers either to figure or caption.

Line 233: I suggest rephrasing to avoid confusion: "(…) and only for shorter lead times. "

Line 242 – 244: Belongs to discussion/conclusion.

Figures 5 and 6: I suggest changing "Lead 1" etc. to Lead time – 1 months for example, to be consistent with the other plots and text. There were substantial differences in performance for the different layers, why not show wet periods for all layers?

Figure 7: Legend and axis names/labels are missing.

Line 256: (..) as shown in Figure 7b.

Line 257 – 259: Belongs to discussion.

Line 267: tab missing after "(Figure 8d)".

Figure 8: Again, I recommend making adjustments to the figure caption, labels and titles, particularly ensuring consistency in lead time labels across all figures/captions and in the text.

Line 285: "nor".

Line 287 – 288: I suggest: "The seasonal model analysed in this study is (..)"

Line 294 – 295: Replace "this paper" with this research/this study or similar.

Line 298: regions

Line 299: "(…), even when considering only the deepest layer; "

Line 301: regions

Line 302: coefficients

Line 301 – 302: Which regions?

Line 304: forecasts

Line 305: regions

Line 305 - 307: This statement needs to be softened as this study did not really predict future events, e.g. to: "This indicates (…)".

Line 308 – 309: Which means? This section should not only include a summary of results but for each statement/bullet point a discussion/conclusion is needed.

Line 310 – 311: Same as above. Conclusion needed from this result.

**RC2: Referee #2**

Review comments for "Assessment of seasonal soil moisture forecast over central Mediterranean toward groundwater management" by Silvestri et al.

The authors have evaluated SEAS5 soil moisture forecast skills in the Mediterranean region. They found that the deepest soil layer (289 cm) has more skill than the upper soil layers. Hence,

the authors conclude that deep soil layer forecast can be potentially valuable for groundwater management regionally.

The authors present a comprehensive review of the relevant literature. The manuscript is well written, and the figure quality is generally good. A novel contribution of this study is the skill in the deepest soil layer forecast. For these reasons, I liked the manuscript, which should eventually be publishable. I recommend the following revisions for the manuscript.

Dear referee,

We would like to thank you for your valuable input. We will try to address your comments as follows.

- Please show the sensitivity of your findings for the reanalysis data selected, which is highly dependent on the selected model (Kumar et al. 2019). Authors may consider alternative data sources, e.g., GLEAMv3 (Martens et al. 2017) and MERRA2 (Gelaro et al. 2017).

Related to the above point, I am somewhat supervised in seeing a lower skill in the upper layer soil moisture anomalies even at shorter lead times, e.g., 1 and 3 months (Figure 2). The root zone (0-1m) has a memory time scale ranging from 2-4 months; I was expecting a higher skill at the shorter lead times.

Additionally, it is unclear if the Authors used ERA5-Land soil moisture data for observations. There are clear differences between ERA5 and ERA5-Land soil moisture data, especially for deeper soil layers (Muñoz-Sabater et al. 2021).

We will definitely use ERA5-Land soil moisture data for observations and we will compare the results with ERA5. However, we decide not to use alternative reanalysis datasets and to maintain a coherence between the soil model used in the seasonal forecasting system and the one used for the production of the reanalysis observation datasets (ERA5 H-TESSEL land surface model) . The comparison of performances between different reanalysis datasets has been done previously [1] for a part of the domain under consideration and ERA5 resulted the best performing dataset in such region. The comparison of different soil models and how they perform in different seasonal forecasting system, or how the findings of this paper depends on the utilized soil model, will be investigated in future work and will be included in the conclusion section. **See Lines 464-466**

[1] Cerlini, P. B., Silvestri, L., Meniconi, S., & Brunone, B. (2023). Performance of three reanalyses in simulating the water table elevation in different shallow unconfined aquifers in Central Italy. *Meteorological Applications*, 30(2), e2118. https://doi.org/10.1002/met.2118

- Process level understanding – please discuss biophysical reasons behind more skillful deeper layer soil moisture prediction in SEAS5. The authors may consider showing the memory time scale in each soil layer for the reanalysis of data and comparing the memory-based predictions with SEAS5 predictions.

We thank referee #2 for his/her useful suggestion. We would like to perform such comparison and we will include the memory time scale in each soil layer, as already done in past works [1], by using a cross-correlation function. However we ask if she/he could indicate us some reference literature or reference method to follow in order to perform and implement a memory-based prediction. **See new section 4.1 and discussion in section 6**

Cerlini, P. B., L. Silvestri, S. Meniconi, and B. Brunone, 2021: Simulation of the Water Table Elevation in Shallow Unconfined Aquifers by means of the ERA5 Soil Moisture Dataset: The Umbria Region Case Study. Earth Interact., 25, 15–32, https://doi.org/10.1175/EI-D-20-0011.1.

- Figure 7 needs a thorough revision – the legend text is missing. X-axis labels are missing. Also, I would suggest two groundwater well data separate in (a) and (b), and they can be compared with the corresponding reanalysis data.

We think that this maybe an error of the printer since all the labels of Figure 7 are visible in the pdf on the screen. Anyway, we will ensure that the figure will be printed correctly and we will follow the suggestion of separating the well data in two figures.

Detailed comments:

Title: 'toward' -> 'for'

The title will be changed accordingly

Line 5: ERA5 reanalysis -> Is this ERA5 or ERA5-Land?

In the original paper it was ERA5, but we will use ERA5-Land in the revised version, as requested by the referee

Line 6-7: 'good performance in the … deepest layer' -> why?

Thanks to the referee suggestion about the memory time scale and to the suggestions of referee #3 about investigating the monthly soil moisture variations, we will try to answer this question and we will include results in the abstract.

Line 165 to 170: SSMA -> Eq1.  -> Add parentheses in the numerator.

This will be corrected

A related comment is that forecast biases (drifts) are a function of the lead time and forecast initialization months (Kumar et al. 2014); it is unclear how you have incorporated these effects in the anomaly calculation. In particular, if you look at Fig. 9 (c), the forecast anomaly does not match the corresponding reanalysis data, even at the start of the forecast! Is it the effect of forecast drifts? If so, this can be easily removed using lead month and forecast initialization month-dependent climatology (Kumar et al. 2014).

We thank the reviewer suggestion. The bias-adjustment method used in the study already considered lead month and month-dependent climatology. **See improved explanation in section methods.**

We will apply all the below suggestions in the revised version of the paper.

Line 204: 'this can be reconducted…' -> what does 'reconducted' refer to. Please consider simplifying this sentence.

Line 205: 'temporal oscillation' -> 'temporal variability'

Figure 1 and other figures, too: please consider using a color-blind-friendly color scheme. For example, I can not clearly distinguish between red and green colors. Additionally, in Figure 1, RMSE < 0.25 looks similar to the color ranging between 0.75 and 1.25.

Figure 2b,c, and f-> why there are stipplings on the Gary color areas; I am assuming they are statistically insignificant correlations.

Figure 2l,m,n -> why darker red areas ( ACC > 0.7) are not stippled, but the yellow area (0.2<ACC<0.4) are stippled.

We double checked the results about the p-values and the correlation coefficient and we also try different types of correlation like the Spearman rank correlation coefficient. However we end up always with the same results. Most of the time (5 %) large correlations coefficient should also have p-values smaller than 0.05 (and viceversa, small correlation coefficients with large p-values). Since the p-value threshold is representing a probability of rejecting the null-hypotesis (no correlation), its meaning is very different from the one of the correlation coefficient, so it is possible to have large correlation coefficients that are not statistically significant and small correlation coefficients that are statistically significant.

Thank you.

References:

- Gelaro, R., and Coauthors, 2017: The Modern-Era Retrospective Analysis for Research and Applications, Version 2 (MERRA-2). *J Climate*, **30,** 5419-5454.

- Kumar, S., P. A. Dirmeyer, and J. Kinter, 2014: Usefulness of ensemble forecasts from NCEP Climate Forecast System in sub-seasonal to intra-annual forecasting. *Geophysical Research Letters*, **41,** 3586-3593.

- Kumar, S., M. Newman, Y. Wang, and B. Livneh, 2019: Potential reemergence of seasonal soil moisture anomalies in North America. *J Climate*, **32,** 2707-2734.

- Martens, B., and Coauthors, 2017: GLEAM v3: satellite-based land evaporation and root-zone soil moisture. *Geosci Model Dev*, **10,** 1903-1925.

- Muñoz-Sabater, J., and Coauthors, 2021: ERA5-Land: A state-of-the-art global reanalysis dataset for land applications. *Earth System Science Data*, **13,** 4349-4383.

We added all the references above.

**RC3: Referee #3**

The main purpose of this study is to determine the predictive power of the SEAS5 system for seasonal soil moisture. The focus was the deepest soil layer at 289 cm in the central Mediterranean region. The accuracy of the SEAS5 (re)forecasts was compared against ERA5 reanalysis datasets, assuming that the ERA5 reflects realistic soil moisture conditions. The specific research question of the study was to predict recharge (stated as a flow toward groundwater – line 88,89) during dry and wet periods.

Improving our forecasting abilities for the water cycle components is a very important subject. The authors tackle this critical problem so that the subject is relevant and timely. The paper is well written. However, the main drawback of this manuscript is that the results were not adequately explained and discussed, leaving many unanswered questions. Therefore, the paper needs to be thoroughly revised, along with the inclusion of additional sections.

Dear referee,

We thank you for your suggestions. We will try to answer them in the revised version of the paper as follows:

My main comments are listed below:

- One of this study's findings is that the forecasted and simulated soil moisture values within the lowest layers were found to be higher than those in the other layers. This is an interesting finding, but the potential reason remains unexplained. Is it because the soil moisture variations in the deeper layers are significantly less than in the surface layers? I would be interested in seeing the monthly soil moisture variations in these deep soil layers.

We thanks the referee for her/his useful suggestions. We will examine both the monthly soil moisture variations and the memory time scale of each layer in the revised paper to investigate our findings more in detail. **See new section 4.1 and discussion in section 6**

- Extensive agricultural activities exist in all the examined regions (Veneto, Umbria, and Naples). However, irrigation was not mentioned in the manuscript. Considering that irrigation may significantly impact both soil moisture and groundwater levels, providing an explanation of the potential implications of irrigation for soil moisture and groundwater level changes might be useful.

- Moreover, in section 2.4, it was stated that groundwater observations are used as a direct proxy to differentiate dry and wet events. However, the dry and wet periods may be observed in water table levels within different time frames. For example, dry periods may have more immediate consequences as the water table declines due to direct water extraction from the aquifers for irrigation purposes. On the other hand, the water table may have a more muted response to wet periods due to the slow vertical movement of soil moisture. While water table observations include these signals, neither ERA5 nor SEAS5 account for irrigation and only indirectly account for them as a result of data assimilation. Further clarification is needed on such connections and the implications of irrigation on the findings of the study.

    We thanks the reviewer for such observations about the influence of irrigation. **See Lines 448-464 in the Conclusions and Discussion section.**

- Although it is important that ERA5 and SEAS5 are independent estimates (i.e., using different initial conditions, data assimilation methods, etc, as stated in lines 137-139), using different soil parameters might lead to very different soil moisture results even though all other forcings are comparable. Please compare the soil hydraulic parameter distributions of both model approaches and explain the potential implications of any existing differences.

- Moreover, in section 3 (line 165), it was stated that both datasets are interpolated over a common resolution (0.25 degrees). However, the way in which this resolution change was handled is missing. Were the soil parameters accounted for during the interpolation? The same moisture amount may lead to different volumetric water contents in different soil textures.

SEAS5 is based on cycle 43r1 of the Integrated Forecasting System of ECMWF. The land surface model of such system (H-TESSEL) does not differ from the one used by the ERA5 reanalysis (IFS cycle 42r1). Since the horizontal grid resolution is very similar (31 km of ERA5 against the 36 km of SEAS5) we expect that both system are consistent with respect to the distribution of soil hydraulic parameters across the entire domain. However the influence of the difference of soil hydraulic parameters (and eventually the effect of interpolation on the ERA5 grid) will be investigated in the revised paper, whenever the soil type is made available by

Copernicus for the SEAS5 system. Moreover, a deeper explanation on similarities and differences between SEAS5 and ERA5 would be included in the Data section. **See new Figure 1 and added reference for soil type interpolation on Lines 145-146.**

- There is a disconnection between the main objective stated at the beginning of the manuscript and the findings of this study. The main research question is, "Can seasonal soil moisture forecasts be used to predict the flow toward groundwater?" However, the study did not attempt to predict groundwater recharge; rather, it sought to find the relationship between soil moisture trends and groundwater level changes in dry and wet periods. Please reword the main objective of the paper.

We thank the reviewer for his/her suggestion. Indeed this paper is a step before the prediction of the flow toward groundwater. We will used the suggestion by the referee in order to modify the research question as "to investigate the relationship between soil moisture trends and groundwater level changes in dry and wet periods". **See new title and Lines 101-104**

- Finally, since both the title and the main objective mention managing water resources, I was expecting some discussion about how this study's findings can be utilized for water management purposes, but such a discussion is missing.

We agree with the referee. As also asked by referee #1, we will modify the conclusions in order to introduce some discussion about how the present study can be used for water management purposes. **See new section Conclusion and Discussion**

All the minor changes below will be corrected in the revised paper.

Minor comments:

- Please include an explanation of dotted areas in the Figure 3 caption.
- Equation 1 is incorrect. Please add a parenthesis to the numerator to fix it.
- Figure 7 is missing legend and axis information.

---

## Referee Report (RR1)

**Re-review of "Assessment of seasonal soil moisture forecasts over Central Mediterranean" by Silvestri et al.**

I have carefully reviewed the authors' responses to my earlier comments, which I find to be generally satisfactory. The inclusion of ERA5-Land data and the added memory analysis (Fig. 3) have enriched the manuscript, making it more comprehensive. Additionally, the expanded discussion section provides valuable insights. Below, I outline three comments that the authors may wish to consider incorporating into the final version of the manuscript:

1. **Memory-Based Prediction Model Reference:** In one of their responses, the authors requested a reference for the memory-based prediction model. The signal component in such a model can be represented as follows:

$$S(t + \tau) = S(t) * \rho(\tau)$$

Here, $S(t + \tau)$ represents the predicted soil moisture anomaly at a lead time of $\tau$, $S(t)$ is the initial condition soil moisture anomaly, and $\rho(\tau)$ is the autocorrelation value at lag time $\tau$. The authors might consider calculating the autocorrelation values using ERA5-Land data and using initial condition anomalies derived from the SEAS5 system to generate a memory-based forecast.

For example:

   o   If for January 2010, S(t)=1.5
   o   And the 6-month lag autocorrelation from ERA5-Land data is 0.60
   o   Then the memory-based prediction at 6-month lead time, i.e., for June 2010 is 1.5·* 0.6=0.9

For reference, the authors might consult Supplementary Fig. 3 in Esit et al. (2021).

2. **Clarification of Ground-Well Data Comparison (Figure 14):** It appears that one of my earlier comments was not fully addressed, possibly due to a misunderstanding. Specifically, my comment referred to "two ground-well data separate in (a) and (b)." The updated Figure 14 does not reflect this separation.

To clarify:

   o   In plot (a), could the authors compare ground water well data at **Umbria** with the corresponding ERA5-Land reanalysis?
   o   Similarly, in plot (b), could the authors compare ground water well data at **Veneto** with the corresponding ERA5-Land reanalysis?

This modification would provide a clearer, location-specific analysis, improving the interpretability of the results.

3. **Rebound in Autocorrelation (Figure 3):** Based on the new Figure 3, I notice an interesting feature in the autocorrelation structure: after an initial decay (as expected), the autocorrelation shows a rebound, reaching a secondary statistically significant maximum at a lag of approximately 300–350 days. This phenomenon merits further exploration.
   o Is this rebound indicative of a seasonal cycle, or could it signify the reemergence of soil moisture anomalies, as hypothesized by Kumar et al. (2019)?
   o A brief discussion of this aspect in the manuscript would be valuable for readers.

Furthermore, given the observed rebound in autocorrelation, the authors might consider exploring the feasibility of longer lead-time forecasts (e.g., a 12-month lead forecast), which could be highly impactful for applications.
* * *
Reference

Esit, M., Kumar, S., Pandey, A., Lawrence, D. M., Rangwala, I., & Yeager, S. (2021). Seasonal to multi-year soil moisture drought forecasting. *npj Climate and Atmospheric Science*, *4*(1), 16.

Kumar, S., Newman, M., Wang, Y., & Livneh, B. (2019). Potential reemergence of seasonal soil moisture anomalies in North America. *Journal of Climate*, *32*(10), 2707-2734.

---

## Author Response (AR2)

"Assessment of seasonal soil moisture forecasts over Central Mediterranean" by Silvestri et al. submitted to Hydrology and Earth System Sciences

Replies to Editor and Referees

**EDITOR (Nunzio Romano)**

Dear Authors,
The revised version of your article has been reviewed by the previous three reviewers, who expressed appreciation for the changes made, even agreeing with some of the somewhat more critical comments. As a reviewer also pointed out, perhaps due to the less experience of some of the group members, it was not always easy to identify the changes made. This was partly due to some lack of clarity regarding which line number was being referred to (whether that of the original article or the revised one).
In general, your revised version was rated favorably for its scientific significance and quality, as well as for the way in which the various elements of the study have been presented.
However, some modifications are still required. The paper is thus released under minor revisions allowing for some additional comments and suggestions received by the reviewers.
I look forward to receiving a new revised version together with the point-by-point replies to the additional comments and suggestions received so far. Should you disagree with a reviewer's comment, please explain why clearly.

Dear Editor,

We thank the Editor for the attention he paid to our paper. We feel the need to apologize for the imprecise reply to the Reviewer's request. Then in the present reply we indicate line numbers for each change (see also tracked-changed version attached). As shown below, in the revised version, we try to address all the requests by Reviewer 1 and 2. In the re-revised version we also added the citation of a paper (Saraceni et al. 2024) accepted in the meantime to corroborate the effectiveness of the use of reanalysis data for water resource management.

**REFEREE 1:**

The authors have addressed most of the specific and minor comments raised in the previous review. They have provided additional details regarding the products used and the performance metrics, incorporated information on antecedent moisture content, and introduced several new figures. These revisions have enhanced the overall quality of the manuscript.

We thank Referee 1 for the attention he/she paid to our paper.

A few minor comments for the authors:
• Please consider using color maps that conform to the standards for color blindness etc.

Reply: In the re-revised version we changed the color maps of Figures 10, 11 and 13, as the other figures passed the test executed by one of the authors who is color blind.

• Most figures lack details regarding the soil layer depths; this information should be included either within the figures or their captions.

Reply: In the re-revised version we add more details of layer depth in all the figure captions.

**REFEREE 2:**

I have carefully reviewed the authors' responses to my earlier comments, which I find to be generally satisfactory. The inclusion of ERA5-Land data and the added memory analysis (Fig. 3) have enriched the manuscript, making it more comprehensive. Additionally, the expanded discussion section provides valuable insights.

Reply: We thank Referee 2 for the attention he/she paid to our paper.

Below, I outline three comments that the authors may wish to consider incorporating into the final version of the manuscript:

1. **Memory-Based Prediction Model Reference**: In one of their responses, the authors requested a reference for the memory-based prediction model. The signal component in such a model can be represented as follows:

    $S(t + \tau) = S(t) * \rho(\tau)$

    Here, $S(t + \tau)$ represents the predicted soil moisture anomaly at a lead time of $\tau$, $S(t)$ is the initial condition soil moisture anomaly, and $\rho(\tau)$ is the autocorrelation value at lag time $\tau$. The authors might consider calculating the autocorrelation values using ERA5- Land data and using initial condition anomalies derived from the SEAS5 system to generate a memory-based forecast. For example:

    - If for January 2010, S(t)=1.5
    - And the 6-month lag autocorrelation from ERA5-Land data is 0.60
    - Then the memory-based prediction at 6-month lead time, i.e., for June 2010 is 1.5·* 0.6=0.9

    For reference, the authors might consult Supplementary Fig. 3 in Esit et al. (2021).

    Reply: We agree with Referee 2 for the relevance of the memory-based prediction model. However, the exploration of the performance of this approach and its comparison with the one used in this paper merits a detailed analysis and it will be considered in future research. In the re-revised version, we add the following sentence in the Discussion to point out this need:

    Lines 472-475: "The second path is to analyze in more detail the behavior of the autocorrelation function of soil moisture anomalies across different soil layers. This to evaluate the role of the seasonal cycle and the reemergence of soil moisture as hypothesized by Kumar et al. (2019). Then, this would allow us to compare the seasonal forecast performance with those obtained by memory-prediction models, following the approach proposed by Esit et al. (2021).

2. **Clarification of Ground-Well Data Comparison (Figure 14)**: It appears that one of my earlier comments was not fully addressed, possibly due to a misunderstanding. Specifically, my comment referred to "two ground-well data separate in (a) and (b)." The updated Figure 14 does not reflect this separation. To clarify:

- In plot (a), could the authors compare ground water well data at Umbria with the corresponding ERA5-Land reanalysis?
- Similarly, in plot (b), could the authors compare ground water well data at Veneto with the corresponding ERA5-Land reanalysis?

This modification would provide a clearer, location-specific analysis, improving the interpretability of the results.

Reply: Done.

3. **Rebound in Autocorrelation (Figure 3):** Based on the new Figure 3, I notice an interesting feature in the autocorrelation structure: after an initial decay (as expected), the autocorrelation shows a rebound, reaching a secondary statistically significant maximum at a lag of approximately 300–350 days. This phenomenon merits further exploration.
   - Is this rebound indicative of a seasonal cycle, or could it signify the reemergence of soil moisture anomalies, as hypothesized by Kumar et al. (2019)?
   - A brief discussion of this aspect in the manuscript would be valuable for readers.

Furthermore, given the observed rebound in autocorrelation, the authors might consider exploring the feasibility of longer lead-time forecasts (e.g., a 12-month lead forecast), which could be highly impactful for applications.

Reply: We thank Reviewer for the precious suggestion. To address his/her request we explore the obtained results with respect to the autocorrelation behavior in all the grid points. As the results do not indicate a unique behavior in all the grid points, we decided to address such behavior in the continuation of our research, as outlined at the end of the discussion (Lines 472-475). Moreover, we added the following lines to the re-revised version:

Lines 226-230: "A further interesting feature is pointed out by the autocorrelation structure. Precisely, after an initial decay (as expected), the autocorrelation shows a rebound with a secondary statistically significant maximum at a lag of approximately 300-350 days. Such rebound could be indicative either of the seasonal cycle or of the reemergence of soil moisture anomalies as hypothesized by Kumar et al. 2019. However, this behavior is not representative of all regions and then merits further explorations in future research."

References

Esit, M., Kumar, S., Pandey, A., Lawrence, D. M., Rangwala, I., & Yeager, S. (2021). Seasonal to multi-year soil moisture drought forecasting. npj Climate and Atmospheric Science, 4(1), 16.

Kumar, S., Newman, M., Wang, Y., & Livneh, B. (2019). Potential reemergence of seasonal soil moisture anomalies in North America. Journal of Climate, 32(10), 2707-2734.

**REFEREE 3:**

Accepted as is (no comments)

Reply: We thank Referee 3 for the attention he/she paid to our paper.